# Light-at-night exposure affects brain development through pineal allopregnanolone-dependent mechanisms

**Shogo Haraguchi[1,2]\*, Masaki Kamata[1], Takuma Tokita[1], Kei-ichiro Tashiro[1], Miku Sato[1], Mitsuki Nozaki[1], Mayumi Okamoto-Katsuyama[3†], Isao Shimizu[3‡], Guofeng Han[4], Vishwajit Sur Chowdhury[4], Xiao-Feng Lei[2], Takuro Miyazaki[2], Joo-ri Kim-Kaneyama[2], Tomoya Nakamachi[5], Kouhei Matsuda[5], Hirokazu Ohtaki[6], Toshinobu Tokumoto[7], Tetsuya Tachibana[8], Akira Miyazaki[2], Kazuyoshi Tsutsui[1]\***

[1]Laboratory of Integrative Brain Sciences, Department of Biology, Waseda University, Tokyo, Japan; [2]Department of Biochemistry, Showa University School of Medicine, Tokyo, Japan; [3]Department of Applied Chemistry, School of Science and Engineering, Waseda University, Tokyo, Japan; [4]Laboratory of Stress Physiology and Metabolism, Graduate School of Bioresource and Bioenvironmental Science, Kyushu University, Fukuoka, Japan; [5]Laboratory of Regulatory Biology, Graduate School of Science and Engineering, University of Toyama, Toyama, Japan; [6]Department of Anatomy, Showa University School of Medicine, Tokyo, Japan; [7]Integrated Bioscience Section, Graduate School of Science and Technology, Shizuoka University, Shizuoka, Japan; [8]Department of Agrobiological Science, Faculty of Agriculture, Ehime University, Matsuyama, Japan

**\*For correspondence:**
shogo.haraguchi@gmail.com (SH);
k-tsutsui@waseda.jp (KT)

**Present address:** [†]School of Life Science and Technology, Tokyo Institute of Technology, Yokohama, Japan

[‡]Deceased 30 October 2014

**Competing interests:** The authors declare that no competing interests exist.

**Abstract** The molecular mechanisms by which environmental light conditions affect cerebellar development are incompletely understood. We showed that circadian disruption by light-at-night induced Purkinje cell death through pineal allopregnanolone (ALLO) activity during early life in chicks. Light-at-night caused the loss of diurnal variation of pineal ALLO synthesis during early life and led to cerebellar Purkinje cell death, which was suppressed by a daily injection of ALLO. The loss of diurnal variation of pineal ALLO synthesis induced not only reduction in pituitary adenylate cyclase-activating polypeptide (PACAP), a neuroprotective hormone, but also transcriptional repression of the cerebellar *Adcyap1* gene that produces PACAP, with subsequent Purkinje cell death. Taken together, pineal ALLO mediated the effect of light on early cerebellar development in chicks.
DOI: https://doi.org/10.7554/eLife.45306.001

## Introduction

Environmental stimuli (*e.g.*, light–dark cycle, temperature, or nutrition) influence the development of plants, animals, and humans. Especially, the light–dark cycle strongly affects development. Several studies have reported that circadian disruption by light-at-night affects weight gain in vertebrates during early neonatal or posthatch life (*Brandon et al., 2002*; *Mann et al., 1986*; *Rozenboim et al., 2013*; *Takahashi et al., 2016*; *Yang et al., 2015*). In addition, previous studies have demonstrated that a light–dark cycle promotes better brain development than does constant light or constant darkness (*Bakkum et al., 1991*; *Brooks et al., 2011*; *Dulcis and Spitzer, 2008*; *Li et al., 2012*; *Ohta et al., 2006*). However, little is known about the molecular mechanisms that control how environmental light conditions affect brain development.

The neuroendocrine system has a critical role in brain development in vertebrates, and its disruption induces abnormal development (*Gore, 2008*; *León-Olea et al., 2014*; *Walker and Gore, 2017*). In the cerebellum, neuroestrogen promotes Purkinje dendritic growth, spinogenesis, and synaptogenesis during neonatal life (*Haraguchi et al., 2012a*; *Sakamoto et al., 2003*; *Sasahara et al., 2007*). The pituitary adenylate cyclase-activating polypeptide (PACAP) protects cerebellar granule cells from apoptosis through the inhibition of caspase-3 activity during development (*Falluel-Morel et al., 2005*; *Vaudry et al., 2003*). Thyroid hormones regulate differentiation of neural cells, synaptogenesis, and myelination (*Bernal, 2007*; *Koibuchi and Chin, 2000*; *Pasquini et al., 1967*). Therefore, for normal cerebellar development, the developmental stage-specific action of various hormones is essential.

Multiple studies have suggested a link between light-at-night-induced circadian disruption and disruption of the neuroendocrine system (*Fonken and Nelson, 2014*; *Fonken et al., 2010*). In the brain, disruption of the neuroendocrine system is caused by exposure to light-at-night (*Fonken et al., 2009*; *Navara and Nelson, 2007*; *Reppert and Weaver, 2002*). Thus, previous studies have suggested that exposure to inappropriate lighting during early life disrupts hormone synthesis and causes abnormal development of the brain during early life. However, the molecular mechanisms that modulate the expression of hormones depending on light conditions during early life are still incompletely understood.

Recently, we have demonstrated that the pineal gland, a photosensitive organ, actively produces a variety of steroids in birds (*Haraguchi et al., 2012a*; *Hatori et al., 2011*). Importantly, pineal allopregnanolone (ALLO), a major pineal steroid, has been shown to prevent the death of cerebellar Purkinje cells by suppressing apoptosis in chicks during development (*Haraguchi et al., 2012a*). Thus, we hypothesized that pineal ALLO synthesis depends on environmental light conditions to mediate cerebellar development during early posthatch life.

To test our hypothesis, we investigated whether light stimuli are involved in the development of the cerebellum via pineal ALLO activity using chicks, an excellent model for studying the effects of light stimuli because chicks show a marked response to changing light conditions.

## Results

### Light-at-night-induced disruption of diurnal variation in pineal ALLO synthesis, followed by Purkinje cell death during early posthatch life

To investigate whether light conditions are involved in pineal ALLO synthesis in newly hatched male chicks, the chicks were incubated under either a 12 hr light–12 hr dark (LD) cycle, constant light (LL), or a 12 hr light–12 hr dark cycle followed by exposure to light for 1 hr from zeitgeber time (ZT) 14 (light-at-night) for 1 week. The mRNA expression of 5α-reductase (srd5a), a steroidogenic enzyme catalyzing the formation of ALLO, showed a marked diurnal change and was high during dark times (ZT16) in the pineal glands of LD chicks (*Figure 1a*). In contrast to the LD chicks, the LL chicks showed consistently low srd5a mRNA levels in the pineal glands (*Figure 1b*). The elevation of srd5a mRNA during ZT16 in LD chicks was suppressed by light during ZT16, 1 hr after the light was put out in light-at-night chicks (*Figure 1c*). In addition, the ALLO concentration and synthesis were higher during ZT16 in the pineal glands of LD chicks than during ZT16 in the pineal glands of LL and light-at-night chicks (*Figure 1d,e*).

To investigate whether the disruption of ALLO synthesis by light-at-night is involved in Purkinje cell survival in chicks during early life, male chicks were incubated under either LD, LL, or light-at-night cycles for 1 week, and all groups were housed under the LD cycle for 2, or 9 weeks (*Figure 2a*). Cerebellar anterior lobe is the region that is the most vulnerable to the reduction of ALLO in the cerebellum (*Haraguchi et al., 2012a*). Compared with LD conditions, LL (lobule IV, p=0.0027; lobule V, p=0.0304) and light-at-night (lobule III, p=0.0129; lobule IV, p=0.0011; lobule V, p=0.0136) conditions increased the number of Purkinje cells that expressed active caspase-3 in lobules III-V of cerebellar vermis on posthatch day 7 (P7) male chicks (*Figure 2c,d* and *Figure 2—figure supplement 3*). Active caspase-3 positive cells were also increased in lobule IV of female chicks by light-at-night (*Figure 2—figure supplement 1a*). After 2 weeks of incubation of all groups under an LD cycle, the effects of light conditions during early life on Purkinje cell number were investigated. Compared with LD conditions, LL (lobule III, p=0.0001; lobule IV, p=0.0005) and light-at-night

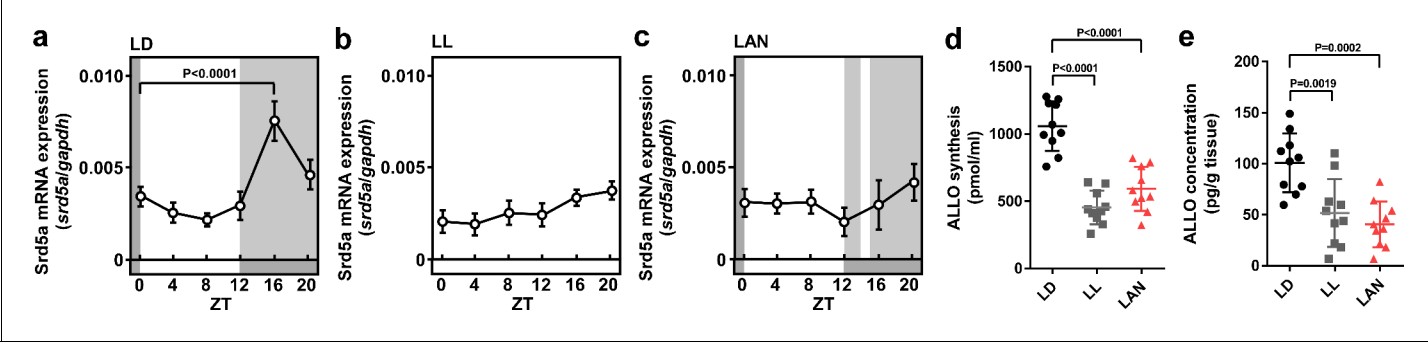

**Figure 1.** Light-at-night-induced disruption of diurnal variation of pineal ALLO synthesis during early life. Diurnal changes in srd5a mRNA expression in the pineal gland under LD (**a**), LL (**b**), or light-at-night (**c**) conditions at P7 chicks (*n* = 10). ALLO synthesis (**d**) and concentration (**e**) in the pineal gland of ZT16 at P7 chicks (*n* = 10). LAN, light-at-night.

DOI: https://doi.org/10.7554/eLife.45306.002

The following source data is available for figure 1:

**Source data 1.** Source data for diurnal changes in srd5a mRNA expression, and ALLO synthesis and concentration in the pineal gland.

DOI: https://doi.org/10.7554/eLife.45306.003

(lobule III, p<0.0001; lobule IV, p<0.0001; lobule V, p=0.0205) conditions decreased the number of Purkinje cells in lobules III-V of cerebellar vermis at P21 chicks (*Figure 2e,f* and *Figure 2—figure supplement 3*). Purkinje cell numbers were also decreased in lobule IV of female chicks by light-at-night (*Figure 2—figure supplement 1b*). In contrast to the number of Purkinje cells, there was no significant difference in the thickness of the molecular layer of Purkinje cells in lobule IV among LD, LL, or light-at-night chicks (*Figure 2—figure supplement 2*). In addition, we investigate whether the decrease in Purkinje cell numbers by light-at-night during the posthatch period persists into adulthood. In adulthood, Purkinje cell numbers were also decreased in lobules III-V of cerebellar vermis by light-at-night during the P1–P7 period (*Figure 2g* and *Figure 2—figure supplement 3*). These results suggest that light-at-night conditions induce disruption of diurnal variation in pineal ALLO and lead to Purkinje cell death in the cerebellum.

## Pineal ALLO prevented Purkinje cell death induced by light-at-night during early life

To investigate whether pineal ALLO prevents Purkinje cell death induced by light-at-night during early life, male chicks were incubated under light-at-night cycle and injected with either ALLO or vehicle-only on a daily basis for 1 week. The chicks were then housed under an LD cycle for 2 weeks (*Figure 3a*). Compared with light-at-night chicks, daily injection of ALLO in light-at-night chicks from P1 to P7 decreased active caspase-3 expression in lobule IV at P7 and improved Purkinje cell survival in lobule IV at P21 (*Figure 3b,c*). These results suggest that pineal ALLO suppresses the apoptosis of Purkinje cells during early life.

## Pineal ALLO prevents apoptosis of Purkinje cells through the mPRα mechanism

To elucidate the mechanism of pineal ALLO action on Purkinje cell survival, we investigated the receptor associated with ALLO action. ALLO acts as an agonist of the γ-aminobutyric acid type A (GABA$_A$) receptor and may act as an agonist of the membrane progesterone receptors α (mPRα), mPRβ, and mPRγ, and the pregnane X receptor (PXR) (*Belelli and Lambert, 2005*; *Frye et al., 2011*; *Langmade et al., 2006*; *Pang et al., 2013*; *Schumacher et al., 2014*). To investigate the expression of the GABA$_A$ receptor, mPRs, and PXR in chick cerebella, RT-PCR analyses were performed. The RT-PCR analyses demonstrated the expression of the α1-subunit of the GABA$_A$ receptor, mPRα, mPRβ, and mPRγ but did not detect the expression of PXR in chick cerebella (*Figure 4a*). Furthermore, to investigate the identified putative receptors that mediate the neuroprotective action of pineal ALLO, we delivered either mPR siRNA or isoallopregnanolone (isoALLO), an antagonist of ALLO, into lobule IV of the cerebellum of newly hatched chicks. Silencing of mPRα increased the

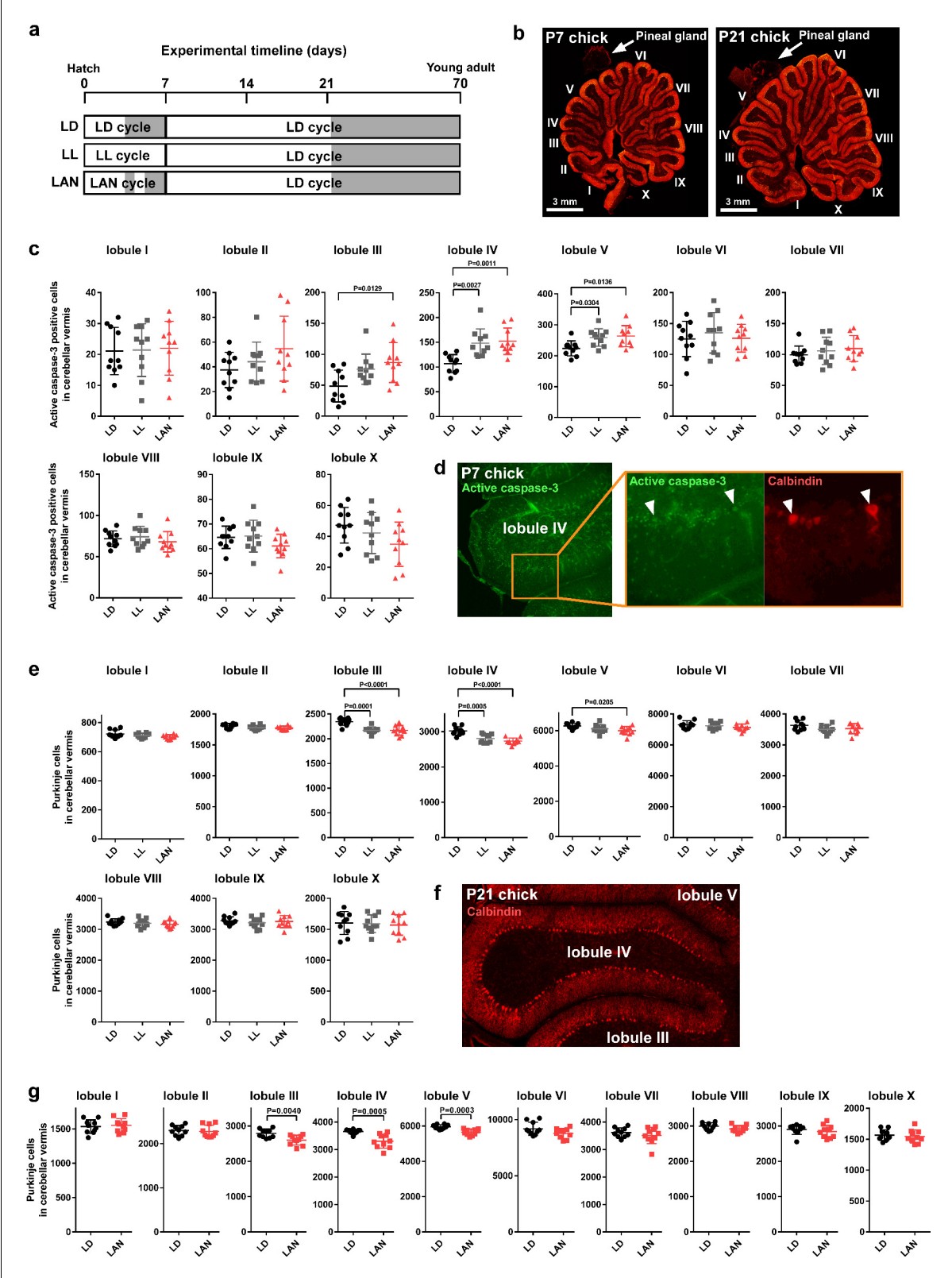

**Figure 2.** Light-at-night-induced Purkinje cell death during early life. (a–g) Male chicks were incubated under LD, LL, or light-at-night cycle for 1 week, and then, all groups were housed under LD cycle for 2 or 9 weeks (a). (b) The lobular structure of the whole cerebellum of chick. Scale bar, 3 mm. (c) Number of Purkinje cells expressing active caspase-3 in each lobule at P7 (*n* = 10). (d) Purkinje cells expressing active caspase-3 in lobule IV at P7. (e)
*Figure 2 continued on next page*

*Figure 2 continued*

Number of Purkinje cells in each lobule at P21 (*n* = 10). (**f**) Purkinje cells in lobule IV at P21. (**g**) Number of Purkinje cells in each lobule at P70 (young adult; *n* = 10). LAN, light-at-night.

DOI: https://doi.org/10.7554/eLife.45306.004

The following source data and figure supplements are available for figure 2:

**Source data 1.** Numbers of active caspase-3 positive cells, and Purkinje cells in male.

DOI: https://doi.org/10.7554/eLife.45306.010

**Figure supplement 1.** Light-at-night-induced Purkinje cell death during early life in female.

DOI: https://doi.org/10.7554/eLife.45306.005

**Figure supplement 1—source data 1.** Numbers of active caspase-3 positive cells, and Purkinje cells in female.

DOI: https://doi.org/10.7554/eLife.45306.006

**Figure supplement 2.** Effect of pineal ALLO on the thickness of the molecular layer.

DOI: https://doi.org/10.7554/eLife.45306.007

**Figure supplement 2—source data 1.** Source data for the thickness of the molecular layer of Purkinje cells.

DOI: https://doi.org/10.7554/eLife.45306.008

**Figure supplement 3.** Estimation plots of the effects of light-at-night on cerebellar Purkinje cells.

DOI: https://doi.org/10.7554/eLife.45306.009

number of Purkinje cells that expressed active caspase-3 in lobule IV of P7 chicks relative to the number in the control siRNA (*Figure 4b*). A daily injection of ALLO did not prevent the activation of caspase-3 in Purkinje cells in mPRα siRNA-transfected chicks (*Figure 4b*). By contrast, transfection of siRNAs for mPRβ and mPRγ into the cerebellum of newly hatched male chicks did not change the expression of active caspase-3 in Purkinje cells from that of the control siRNA-transfected chicks (*Figure 4c,d*). A daily injection of isoALLO did not change the number of Purkinje cells that expressed active caspase-3 in lobule IV of P7 chicks relative to the number in the control chicks (*Figure 4e*). These results suggest that mPRα may function as a receptor for pineal ALLO in the developing cerebellum.

Immunohistochemical (IHC) analysis using anti-goldfish mPRα antiserum was performed to analyze the cellular localization of mPRα in the chick cerebellum. To confirm that this antibody recognizes the chicken mPRα protein, we first performed a western blot analysis on the extracts of COS-7 cells transfected with chicken mPRα cDNA. A single immunoreactive band (approximately 42 kDa) was detected (*Figure 4—figure supplement 1*). When the cerebellum extracts were analyzed, a single band (approximately 42 kDa) was detected at the same position (*Figure 4—figure supplement 1*). Distinct mPRα immunoreactivity was observed in the Purkinje cells (*Figure 4f*).

To investigate the binding of radiolabeled ALLO to chicken mPRα, radioreceptor assays were performed. Saturation analysis and Scatchard plots show the presence of a high-affinity (dissociation constant, 11.8 ± 3.1 nM), limited-capacity (maximal binding capacity, 0.91 ± 0.29 nM), saturable, single-binding site for [$^3$H]-ALLO in the membrane fraction of the COS-7 cells transfected with chicken mPRα cDNA (*Figure 4g*). Furthermore, to test the binding of ALLO to chick mPRα, we developed fluorescein-labeled ALLO (*Figure 4h*). COS-7 cells expressing mPRα were stained with fluorescein-labeled ALLO. Confocal fluorescence microscopy revealed that fluorescein-labeled ALLO bound to COS-7 cells expressing mPRα and that an excess of ALLO competed with fluorescein-labeled ALLO for binding to COS-7 cells expressing mPRα (*Figure 4i* and *Figure 4—figure supplement 2*).

## Pineal ALLO-induced expression of PACAP, an endogenous neuroprotective factor, in cerebellar Purkinje cells

ALLO acts as a neuroprotective factor; however, the molecular mechanisms behind this neuroprotection are currently unknown. To clarify the neuroprotective mechanisms of ALLO in Purkinje cells, we investigated the effect of ALLO on the expression of neuroprotective and neurotoxic factors (*Bernal, 2007*; *Falluel-Morel et al., 2005*; *Haraguchi et al., 2012a*; *Koibuchi and Chin, 2000*; *Pasquini et al., 1967*; *Sakamoto et al., 2003*; *Sasahara et al., 2007*; *Vaudry et al., 2003*) in the whole cerebellum. Pinealectomy (Px) at P1 decreased the expression of PACAP and brain-derived neurotrophic factor (BDNF) mRNAs at P7 (*Figure 5a*). On the other hand, Px did not influence the expression of neurotrophin-3 (NT-3), cytochrome P450 aromatase (P450arom), 3β-hydroxysteroid dehydrogenase (3β-HSD), type II deiodinase (Dio2), and insulin-like growth factor I (IGF-1) mRNAs

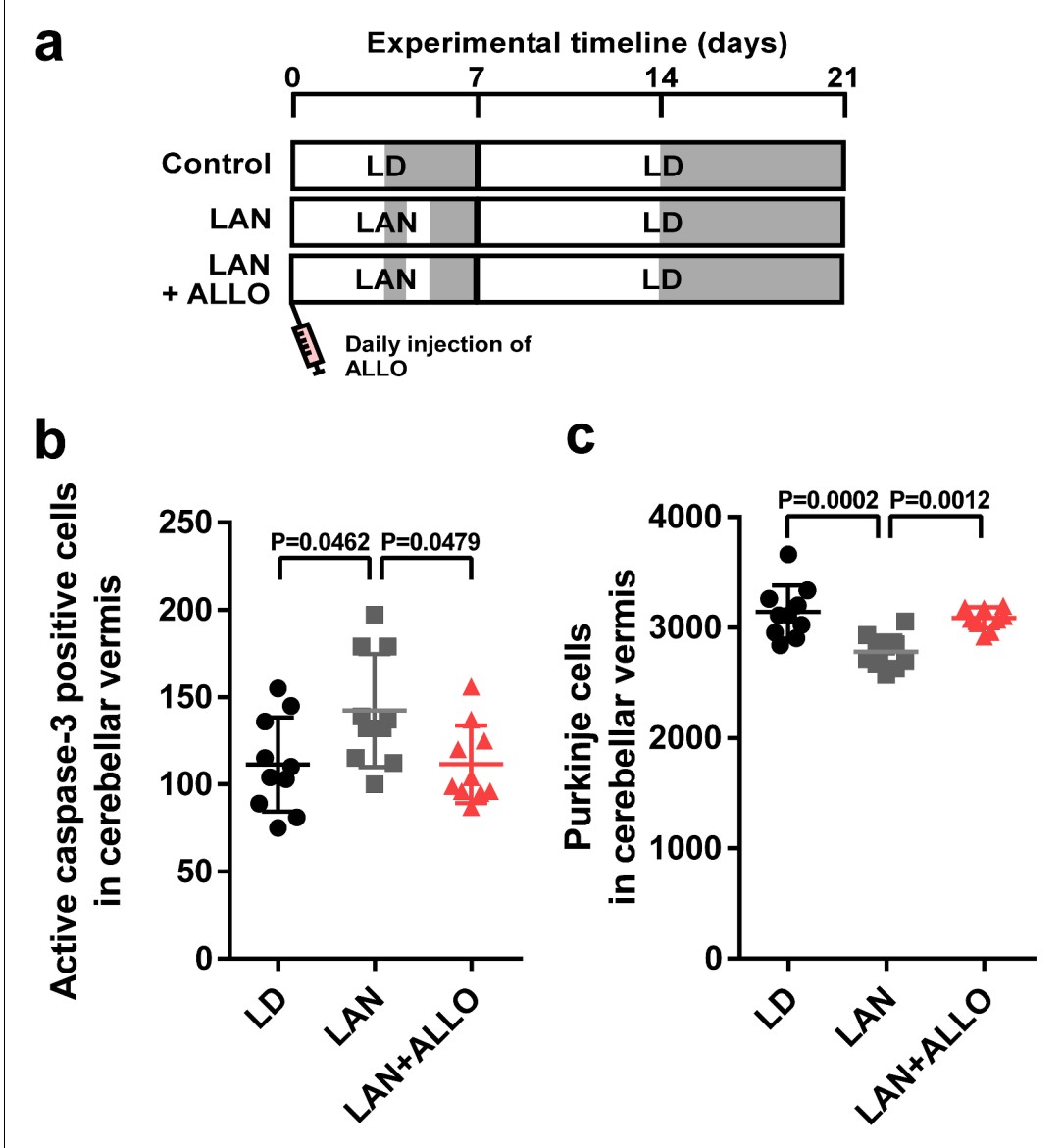

**Figure 3.** Light-at-night-induced Purkinje cell death was rescued by ALLO injection during early life. (**a**) Male chicks were incubated under LD or light-at-night cycle for 1 week, and then, all groups were housed under LD cycle for 2 weeks. Half of the light-at-night chicks were treated with a daily injection of ALLO from P1 to P7. (**b**) Number of Purkinje cells expressing active caspase-3 in lobule IV at P7 (*n* = 10). (**c**) Number of Purkinje cells in lobule IV at P21 (*n* = 10). LAN, light-at-night.

DOI: https://doi.org/10.7554/eLife.45306.011

The following source data is available for figure 3:

**Source data 1.** Numbers of active caspase-3 positive cells, and Purkinje cells in male.
DOI: https://doi.org/10.7554/eLife.45306.012

(*Figure 5a*). In addition, a daily injection of ALLO in Px chicks increased the expression of PACAP relative to that in Px chicks (*Figure 5a*).

To investigate whether light conditions are also involved in the expression of PACAP during early life, male chicks were incubated under either LD or light-at-night conditions for 1 week. The expression of PACAP mRNA showed a marked diurnal change in the cerebellum and was high during dark times (ZT16 and ZT20) in LD chicks (*Figure 5b*). The elevation of PACAP mRNA at ZT16 in LD chicks was suppressed at ZT16 in light-at-night chicks (*Figure 5b*). These changes were in parallel with those in srd5a mRNA in the pineal gland (*Figure 1a,c*). Px increased the number of Purkinje cells

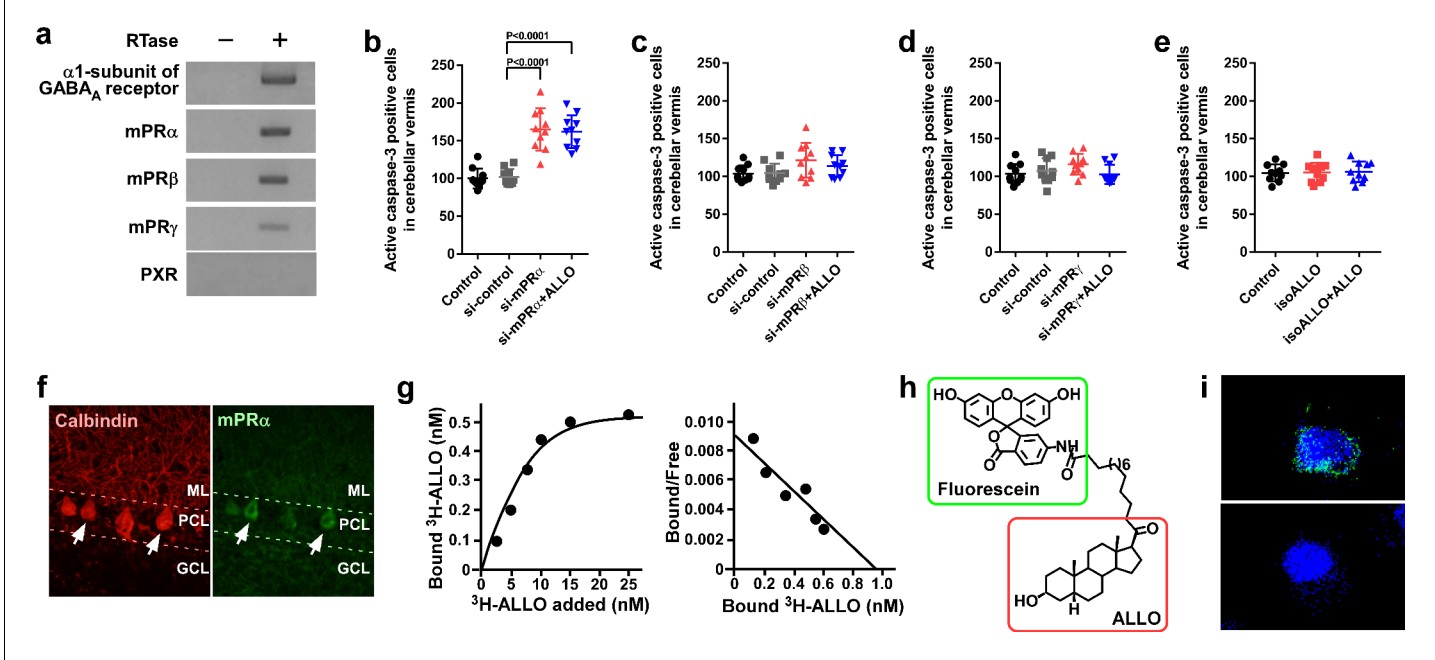

**Figure 4.** Pineal ALLO prevented Purkinje cell death through mPRα in cerebellar Purkinje cells. (**a**) Shown are the expressions of the α1-subunit of the GABA_A receptor, mPRα, mPRβ, mPRγ, and PXR in the cerebellum of P1 chicks (similar results were obtained in repeated experiments using three different samples). (**b–e**) Number of Purkinje cells expressing active caspase-3 in lobule IV at P7 by receptor gene silencing as a candidate of the ALLO receptor in the cerebellar cortex at P1 ($n = 10$). (**f**) Immunohistochemistry of mPRα in the cerebellum of P1 chicks (similar results were obtained in repeated experiments using three different samples). (**g**) Representative saturation curve and Scatchard plot of specific [³H]-ALLO binding to plasma membranes of mPRα transfected cells ($n = 4$). (**h**) Structure of fluorescein-labeled ALLO. (**i**) Confocal images of mPRα-expressing cells stained with fluorescein-labeled ALLO under permeabilized conditions in the absence (upper panel) or presence (lower panel) of excess competing ALLO (similar results were obtained in repeated experiments using three different samples). ML, molecular layer; PCL, Purkinje cell layer; GCL, granule cell layer.
DOI: https://doi.org/10.7554/eLife.45306.013

The following source data and figure supplements are available for figure 4:

**Source data 1.** Numbers of active caspase-3 positive cells in male.
DOI: https://doi.org/10.7554/eLife.45306.016
**Figure supplement 1.** Western blot analysis with the anti-goldfish mPRα antibody.
DOI: https://doi.org/10.7554/eLife.45306.014
**Figure supplement 2.** Synthetic scheme for the synthesis of the Fluorescein-labeled ALLO.
DOI: https://doi.org/10.7554/eLife.45306.015

that expressed active caspase-3 in lobule IV of P7 chicks relative to those in the control (*Figure 5c*). A daily injection of ALLO or PACAP in Px chicks decreased the number of Purkinje cells expressing active caspase-3 in lobule IV at P7 (*Figure 5c*). Px at P1 decreased the number of Purkinje cells in lobule IV at P21 (*Figure 5d*). A daily injection of ALLO or PACAP in Px chicks from P1 to P7 improved Purkinje cell survival in lobule IV at P21 (*Figure 5d*). Clear PACAP and PAC1 immunoreactivities were observed in Purkinje cells (*Figure 5e,f*). These results suggest that PACAP mediates the neuroprotective action of pineal ALLO during early life.

## The neuroprotective actions of pineal ALLO on Purkinje cells are restricted during the first week posthatch

The postnatal development of the cerebellar cortex irreversibly leads to the maturation of the cerebellum, including the maturation of Purkinje cells. For normal cerebellar development, the developmental stage-specific actions of various hormones in the developing cerebellum are essential. Thus, to investigate whether the neuroprotective actions of pineal ALLO on Purkinje cells are a stage-specific role of ALLO during early life, male chicks were housed under LD cycle for 1 week during the first week posthatch and then incubated under the light-at-night cycle and injected with either ALLO or a solvent daily for 1 week during the second week posthatch. Thereafter, the chicks were housed

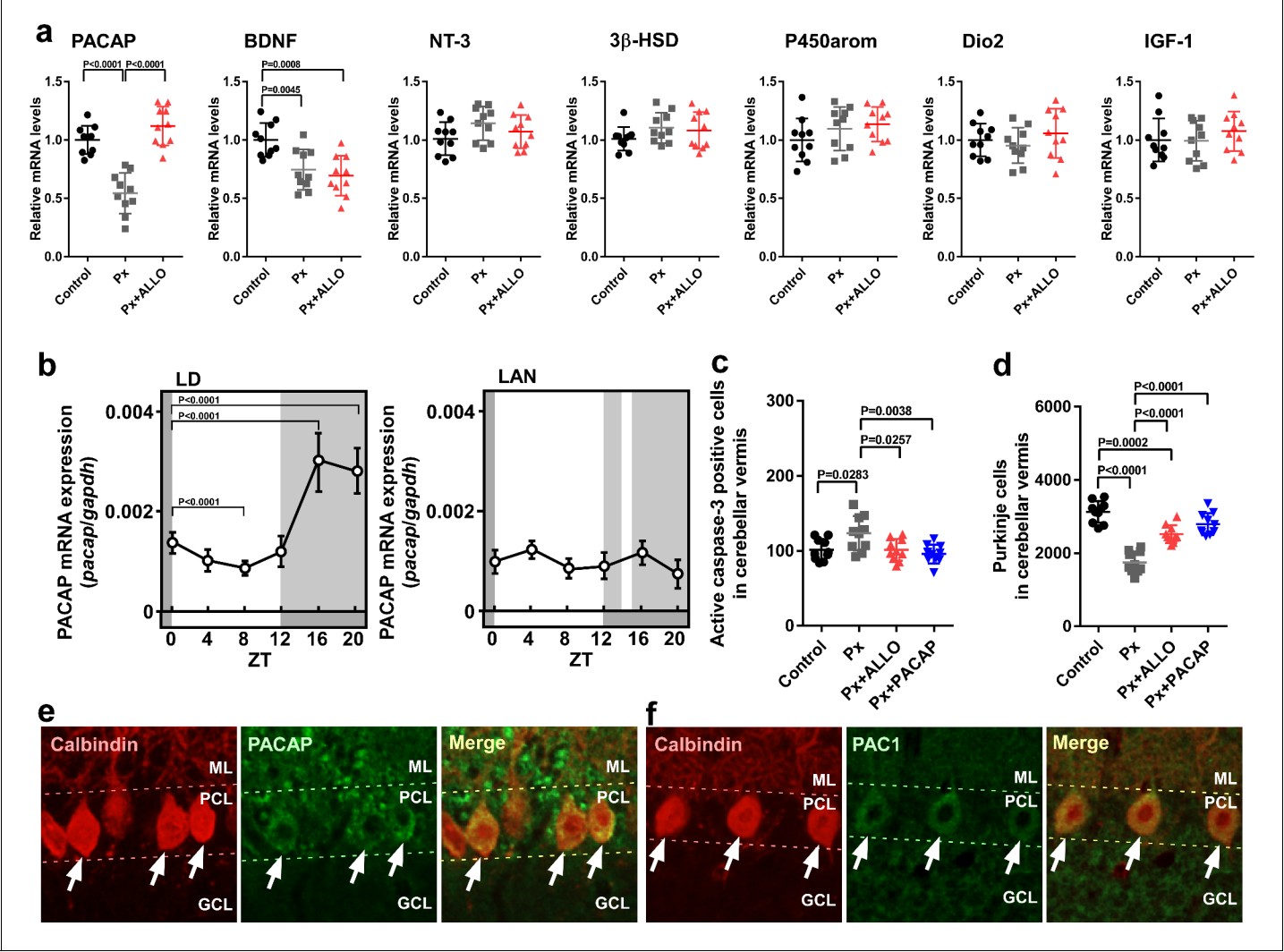

**Figure 5.** Pineal ALLO-induced expression of PACAP, an endogenous neuroprotective factor, in cerebellar Purkinje cells. (a) Shown are the effects of Px or Px plus ALLO on the expression of neuroprotective and neurotoxic factors ($n = 10$). (b) Diurnal changes in PACAP mRNA expression in the cerebellum under LD or light-at-night conditions in P7 chicks ($n = 10$). (c) Number of Purkinje cells expressing active caspase-3 in lobule IV at P7 ($n = 10$). (d) Number of Purkinje cells in lobule IV at P21 ($n = 10$). Immunohistochemistry of PACAP (e) and PAC1 (f) in the cerebellum of P1 chicks (similar results were obtained in repeated experiments using three different samples). Calbindin is used as a marker for Purkinje cells. ML, molecular layer; PCL, Purkinje cell layer; GCL, granule cell layer; LAN, light-at-night.
DOI: https://doi.org/10.7554/eLife.45306.017

The following source data is available for figure 5:

**Source data 1.** source data for mRNA expressions (a), diurnal changes in PACAP mRNA expression (b), and numbers of Purkinje cells (c and d) in male.
DOI: https://doi.org/10.7554/eLife.45306.018

under LD cycle for 1 week during the third week posthatch (*Figure 6a*). Light-at-night chicks showed a decreased Purkinje cell number at P21 relative to that in the control, but daily injections of ALLO in light-at-night chicks from P8 to P14 did not rescue Purkinje cells from death in lobule IV at P21 and, furthermore, did not suppress the expression of active caspase-3 in lobule IV at P14 (*Figure 6b, c*). In summary, the neuroprotective actions of pineal ALLO on Purkinje cells are restricted during the first week posthatch.

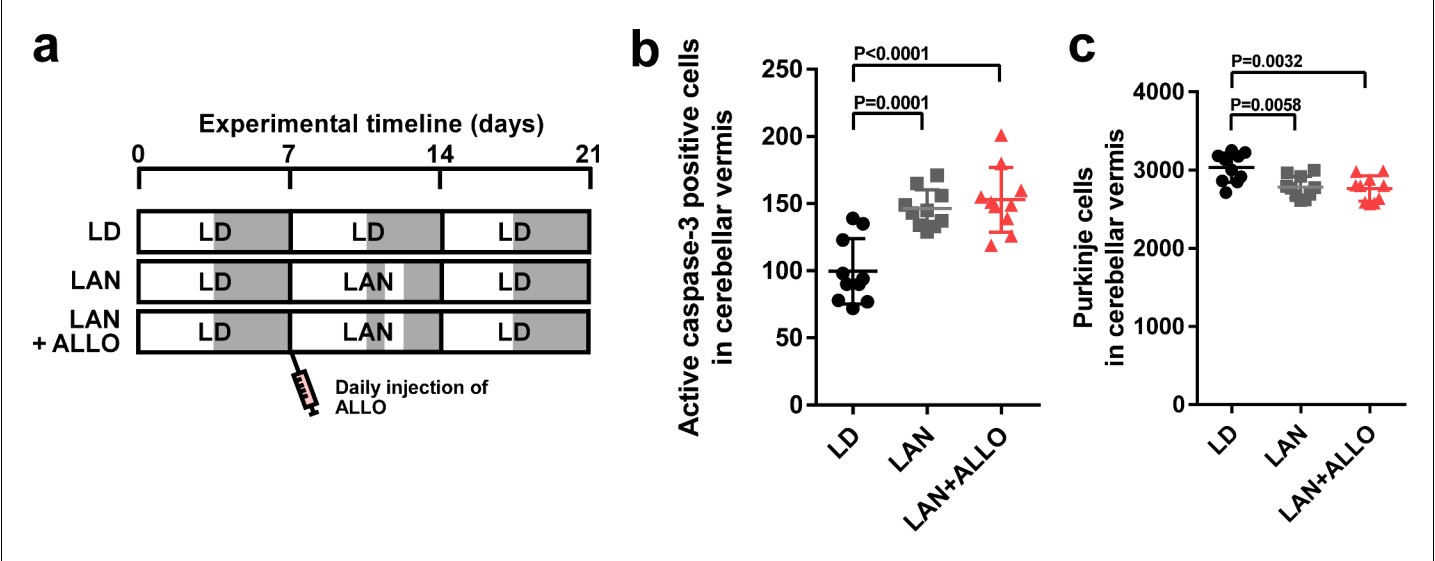

**Figure 6.** ALLO did not rescue Purkinje cells from death during the second week posthatch. Male chicks were housed under LD cycle for 1 week during the first week posthatch. They were then incubated under light-at-night cycle and injected with ALLO or a solvent daily for 1 week during the second week posthatch. Finally, the chicks were housed under LD cycle for a week during the third week posthatch (a). (b) Number of Purkinje cells expressing active caspase-3 in lobule IV at P14 ($n$ = 10). (c) Number of Purkinje cells in lobule IV at P21 ($n$ = 10). LAN, light-at-night.

DOI: https://doi.org/10.7554/eLife.45306.019

The following source data is available for figure 6:

**Source data 1.** Numbers of active caspase-3 positive cells, and Purkinje cells in male.
DOI: https://doi.org/10.7554/eLife.45306.020

## The neuroprotective actions of PACAP on Purkinje cells are not restricted during early life

To investigate whether the neuroprotective actions of PACAP on Purkinje cells are restricted, as were those of pineal ALLO, during early life, male chicks were incubated under light-at-night cycle and injected with either PACAP or solvent daily for 1 week. These chicks were then housed under LD cycle for 2 weeks (*Figure 7a*). Daily injection of PACAP in light-at-night chicks from P1 to P7 decreased active caspase-3 expression in lobule IV at P7 relative to that in light-at-night chicks and improved Purkinje cell survival in lobule IV at P21 (*Figure 7b,c*). Subsequently, male chicks were housed under LD cycle for 1 week during the first week posthatch and then incubated under light-at-night cycle and injected with either PACAP or solvent daily for 1 week during the second week posthatch. Subsequently, these chicks were housed under LD cycle for 1 week during the third week posthatch (*Figure 7d*). Compared with the control, light-at-night chicks were found to have a decreased Purkinje cell number at P21, and daily injection of PACAP in light-at-night chicks from P8 to P14 suppressed the expression of active caspase-3 expression in lobule IV at P14 and rescued Purkinje cells in lobule IV at P21 (*Figure 7e,f*). In summary, the neuroprotective actions of PACAP on Purkinje cells were not restricted during the first week posthatch.

## Pineal ALLO did not induce the expression of PACAP in the cerebellum during the second week posthatch

In terms of the difference between the neuroprotective actions of pineal ALLO and PACAP (*Figures 6* and *7*), we hypothesized that pineal ALLO did not induce the expression of PACAP in the cerebellum during the second week posthatch. To test this hypothesis, ALLO was injected daily from P8 to P14 in Px at P8 chicks. Px at P8 decreased PACAP expression in the cerebellum at P14 relative to that in the control (*Figure 8a*). A daily injection of ALLO in Px at P8 chicks from P8 to P14 did not induce the expression of PACAP mRNA (*Figure 8a*). In addition, we demonstrated that PACAP expression decreased gradually according to the maturation of the cerebellum during early life (*Figure 8b*).

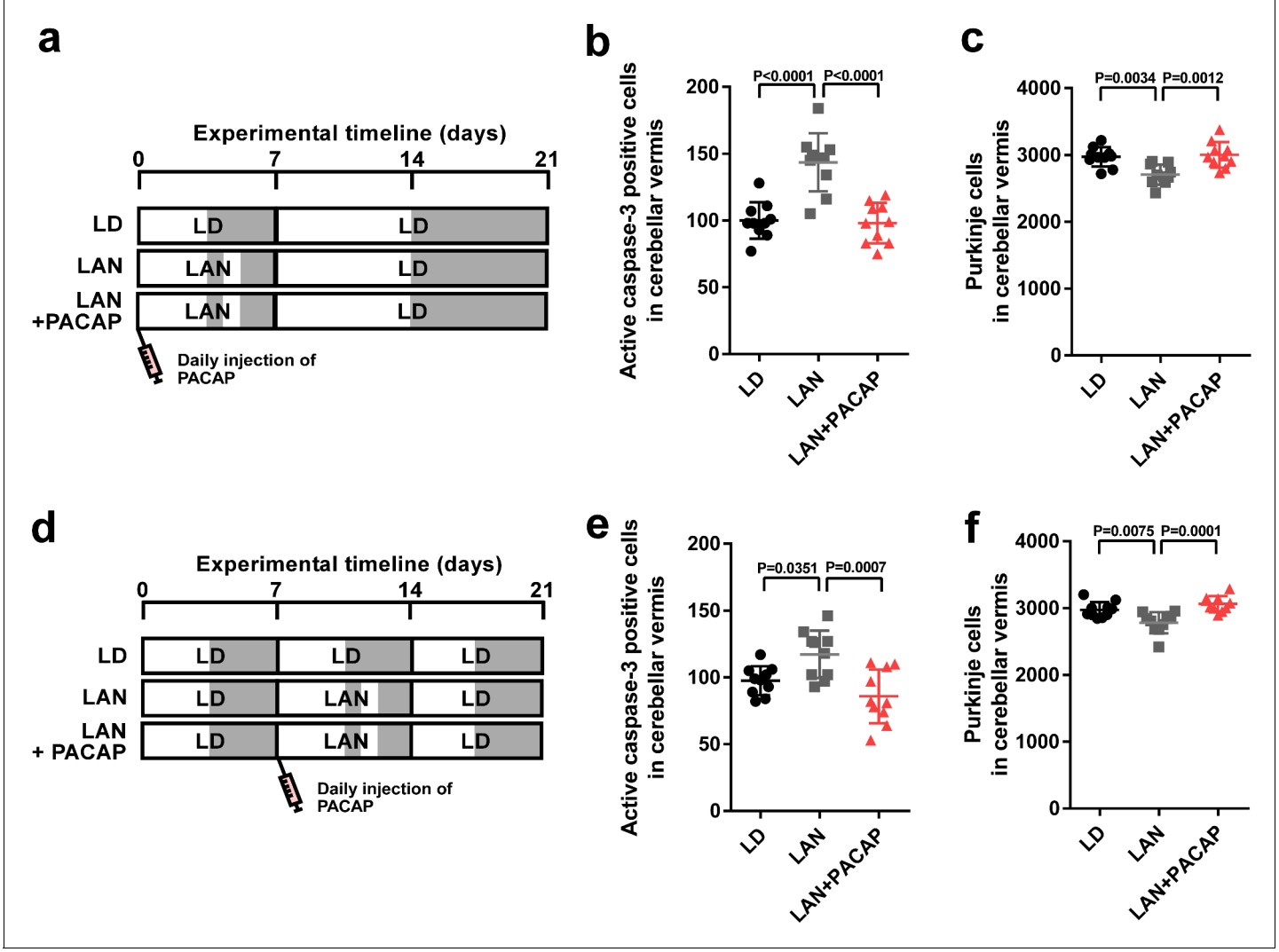

**Figure 7.** Neuroprotective effects of PACAP were not restricted during early life. (**a**) Male chicks were incubated under LD or light-at-night cycle for 1 week during the first week posthatch and injected either with PACAP or solvent daily, and then, all groups were housed under LD cycle for 2 weeks during the second and third weeks posthatch. (**b**) Number of Purkinje cells expressing active caspase-3 in lobule IV at P7 (*n* = 10). (**c**) Number of Purkinje cells in lobule IV at P21 (*n* = 10). Male chicks were housed under LD cycle for 1 week during the first week posthatch and then incubated under light-at-night cycle and injected PACAP or solvent daily for 1 week during the second week posthatch. Finally, chicks were housed under LD cycle for 1 week during the third week posthatch (**d**). (**e**) Number of Purkinje cells expressing active caspase-3 in lobule IV at P14 (*n* = 10). (**f**) Number of Purkinje cells in lobule IV at P21 (*n* = 10). LAN, light-at-night.

DOI: https://doi.org/10.7554/eLife.45306.021

The following source data is available for figure 7:

**Source data 1.** Numbers of active caspase-3 positive cells, and Purkinje cells in male.

DOI: https://doi.org/10.7554/eLife.45306.022

## H3K9me3 levels at the *Adcyap1* gene promoter increased in Purkinje cells according to the maturation of the cerebellum

Histone modification is a major form of epigenetic gene regulation that is critical in many neuronal processes (*Kundakovic and Champagne, 2015*; *Montgomery et al., 2009*; *Toyoda et al., 2014*). Trimethylation of lysine 9 or 27 on histone H3 and lysine 20 on histone H4 (H3K9me3, H3K27me3, and H4K20me3) is associated with transcriptional repression.

Thus, to reveal the molecular mechanisms underlying the difference in actions of pineal ALLO on the expression of PACAP between the first and second weeks after hatch, we investigated histone

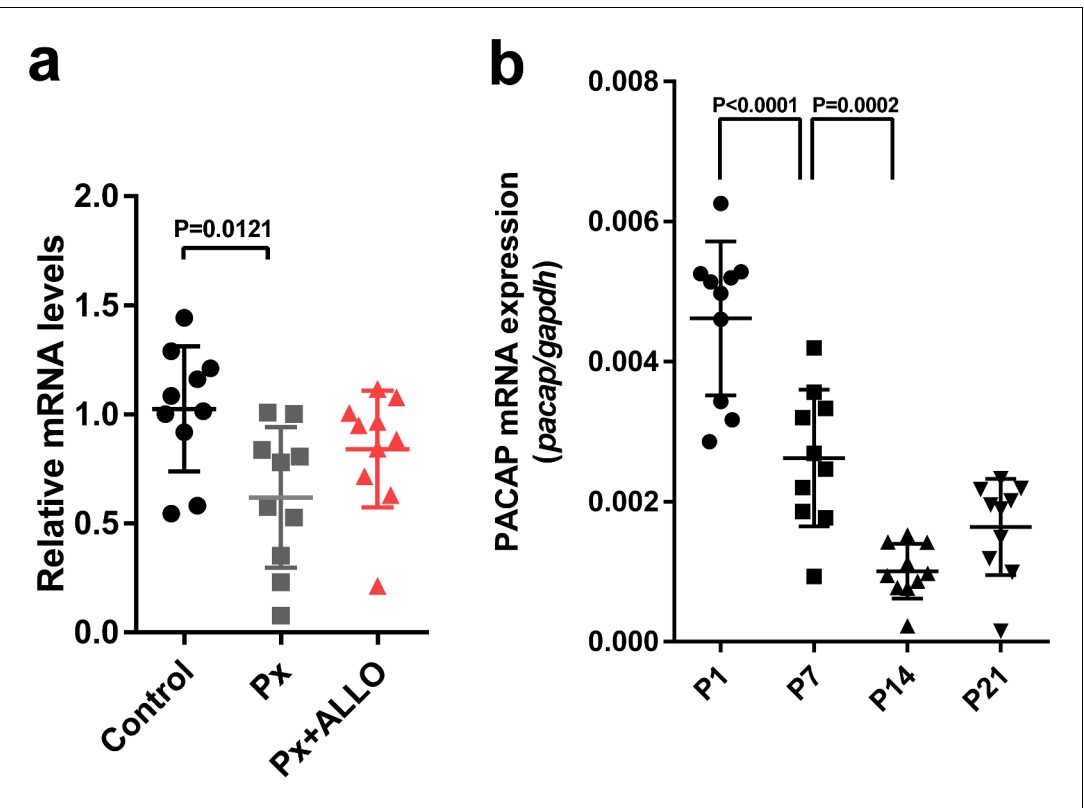

**Figure 8.** Pineal ALLO did not induce the expression of PACAP in the cerebellum during the second week posthatch. (**a**) The effects of Px or Px plus ALLO on the expression of PACAP in the cerebellum during the second week posthatch (*n* = 10). (**b**) Changes in the expression of PACAP during early life.
DOI: https://doi.org/10.7554/eLife.45306.023
The following source data is available for figure 8:

**Source data 1.** Source data for mRNA expressions.
DOI: https://doi.org/10.7554/eLife.45306.024

modifications associated with the transcriptional repression of PACAP. Purkinje cell layer sections were microdissected, and the samples were analyzed by chromatin immunoprecipitation and quantitative PCR (ChIP-qPCR). The association levels of H3K9me3 at the *Adcyap1* gene promoter were higher at P2 than those at P8 in the Purkinje cell layer of the cerebellum (*Figure 9a*), whereas H3K27me3 and H4K20me3 levels did not change between the first and second weeks posthatch (*Figure 9a*). These results suggested that H3K9me3 may have an important role in *Adcyap1* gene expression, so we measured the levels of H3K9me3 in Purkinje cell nuclei. H3K9me3 immunoreactivity in the nuclei of Purkinje cells was measured using ImageJ. The intensity was found to be high at P2 relative to that at P8 in Purkinje cells (*Figure 9b,c*).

## Light-at-night and Px increased H3K9me3 levels at the *Adcyap1* gene promoter

It is known that environmental factors affect the developing brain through epigenetic mechanisms. Various environmental stimuli cause changes in the profile of histone modification (*Roth and Sweatt, 2011*).

Thus, we investigated the effects of light-at-night or ALLO on histone modification in the *Adcyap1* gene promoter. Male chicks were incubated under LD or light-at-night cycle during the first week posthatch, and the resulting histone modifications on the *Adcyap1* gene promoter were investigated. light-at-night was found to increase the levels of H3K9me3 in the *Adcyap1* gene promoter in the Purkinje cell layer of the cerebellum (*Figure 10a*). In addition, a daily injection of ALLO in

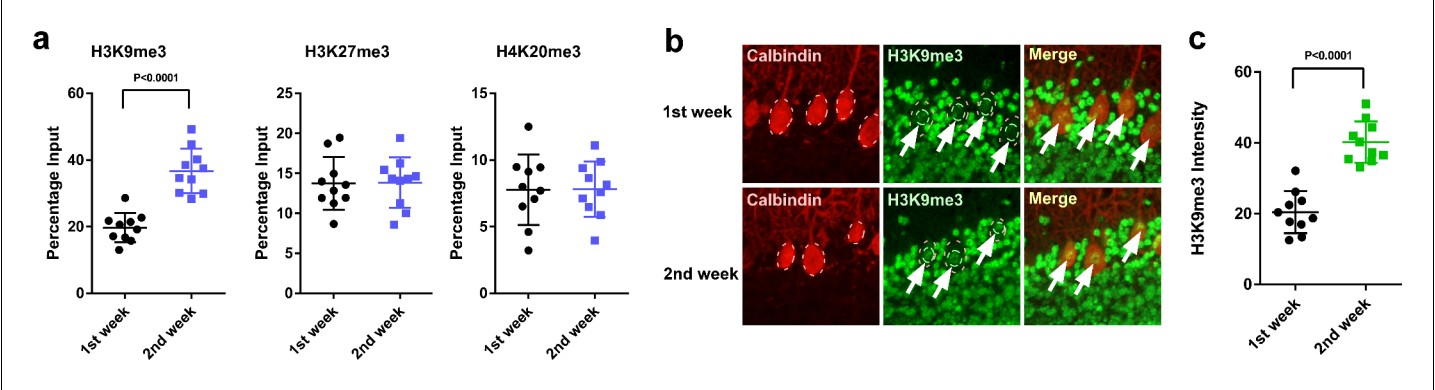

**Figure 9.** H3K9me3 levels at the *Adcyap1* promoter increased in Purkinje cells during development. (**a**) Changes in the levels of histone modification between the first and second weeks of posthatch. (**b**) Immunohistochemistry of H3K9me3 in the cerebellum of P1 chicks. Calbindin is used as a marker for Purkinje cells. (**c**) The relative intensity of H3K9me3 in the nucleus of Purkinje cells (*n* = 10).

DOI: https://doi.org/10.7554/eLife.45306.025

The following source data is available for figure 9:

**Source data 1.** Source data for the levels of histone modification.
DOI: https://doi.org/10.7554/eLife.45306.026

light-at-night chicks from P1 to P7 suppressed methylation of H3K9me3 in the *Adcyap1* gene promoter in the Purkinje cell layer of lobule IV at P7 relative to that in light-at-night chicks (*Figure 10a*).

Px at P1 increased the levels of H3K9me3 at the *Adcyap1* gene promoter in the Purkinje cell layer of lobule IV at P7 (*Figure 10b*). A daily injection of ALLO in Px chicks from P1 to P7 suppressed methylation of H3K9me3 at the *Adcyap1* gene promoter in the Purkinje cell layer of lobule IV at P7 relative to that in Px chicks (*Figure 10b*).

## Discussion

Circadian rhythms are present in almost all plants and animals. However, modern life induces chronic circadian disruption through artificial light and such disruption is associated with a decline in mental and physical health (*Kantermann and Roenneberg, 2009*; *Wu et al., 2011*; *Smarr et al., 2017*). The most potent cue in circadian rhythm disruption is inappropriately timed bright light (*e.g.*, light-at-night). To understand the influences of light-at-night, as a model of modern life induced chronic circadian disruption, on mental and physical health in humans, many studies have been conducted using laboratory mice. However, it is important to bear in mind that laboratory mice are mainly nocturnal animals, while humans are diurnal. Thus, in this study we used birds to provide a diurnal animal model to understand the influence of light-at-night on the development of the cerebellum in mammals including humans.

Autism spectrum disorder is a complex neurodevelopmental disorder. Autism spectrum disorder is characterized by early-onset difficulties in social interaction, repetitive behavior, and verbal and non-verbal communication (*Lai et al., 2014*). A number of neurobiological hypotheses have been put forward to account for autism behaviors that implicate neural and network abnormalities in the cerebellum that include the cerebellar vermis area (*Piochon et al., 2014*). Therefore, chronic circadian disruption by artificial light may involve autism spectrum disorder by decreasing the number of Purkinje cells in the vermis area by light-at-night. Further studies are needed to investigate the connection between a decrease in Purkinje cells caused by light-at-night and autism spectrum disorder.

Light-at-night-induced circadian disruption alters the synthesis and secretion of various hormones, such as melatonin (a pineal hormone) and cortisol (an adrenal steroid hormone), particularly in vertebrates, including humans (*Klein, 2006*). These disruptions of the synthesis and release of hormones by inappropriate timed bright light increase the risk of breast cancer, prostate cancer, obesity, diabetes, and depression (*Kantermann and Roenneberg, 2009*; *Shi et al., 2013*; *Smarr et al., 2017*; *Wu et al., 2011*). Circadian disruptions by bright light have also been shown to affect the health of infants (*Mann et al., 1986*). Inappropriate light conditions reduce weight gain in preterm infants in a

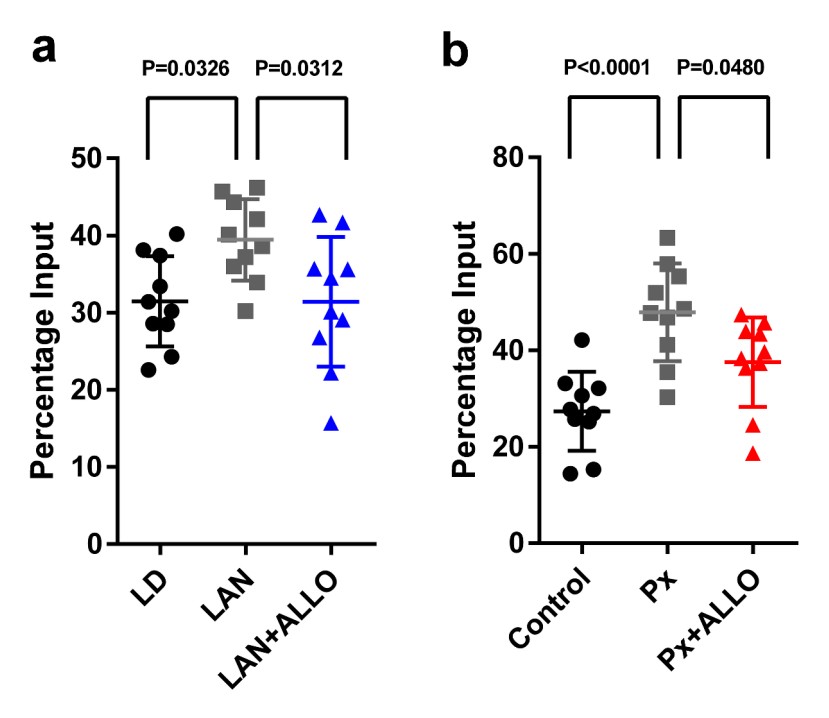

**Figure 10.** Light-at-night and Px increased in H3K9me3 levels at the *Adcyap1* promoter. (**a**) The effects of light-at-night and light-at-night plus ALLO on the level of H3K9me3 at the *Adcyap1* gene promoter (*n* = 10). (**b**) The effects of Px and Px plus ALLO on the level of H3K9me3 at the *Adcyap1* gene promoter (*n* = 10). LAN, light-at-night.

DOI: https://doi.org/10.7554/eLife.45306.027

The following source data is available for figure 10:

**Source data 1.** source data for H3K9me3 levels at the *Adcyap1* promoter.

DOI: https://doi.org/10.7554/eLife.45306.028

newborn nursery via circadian disruption of hormone synthesis (*Brandon et al., 2002*; *Mann et al., 1986*). Bright light can be a stress factor. It has been shown that stress in the early life of rats delays development and accounts for a number of abnormalities in the brain (*Ellenbroek et al., 2005*). As part of the organization process, natural apoptotic cell death occurs in rodents and birds just after birth and around the second postnatal week (*Prakash and Wurst, 2006*). In the human brain, the developmental processes of proliferation and migration are generally finalized after 24 weeks of gestation, but apoptotic cell death, synaptogenesis, and neuronal differentiation take place up until the three year of life (*Huppertz-Kessler et al., 2012*). Thus, previous and present studies have both suggested that inappropriate lighting disrupts hormone synthesis and causes abnormal development of the brain during early life (*Figure 11*).

We previously reported that the pineal gland is an important steroidogenic organ, and pineal ALLO may have an important role in the prevention of Purkinje cell death for the normal development of the cerebellum (*Haraguchi et al., 2012a*; *Hatori et al., 2011*). The pineal gland of vertebrates responds to light via direct and indirect mechanisms (*Klein, 2006*; *Maronde and Stehle, 2007*) and has important roles in the circadian organization of vertebrates. However, the influence of light-at-night on the production of pineal ALLO has yet to be elucidated. In this study, we demonstrated that light-at-night-induced circadian disruption abolished the diurnal rhythm of pineal ALLO synthesis. In addition, the disruption of diurnal variations of pineal ALLO-induced Purkinje cell death in the developing cerebellum of chicks. These results suggest that the disruption of diurnal variations of pineal ALLO causes a constant low level of pineal ALLO synthesis, which induces the death of Purkinje cells during development. A number of studies have found that stage-specific actions of various hormones are essential for normal cerebellar development. For instance, thyroid deficiency during

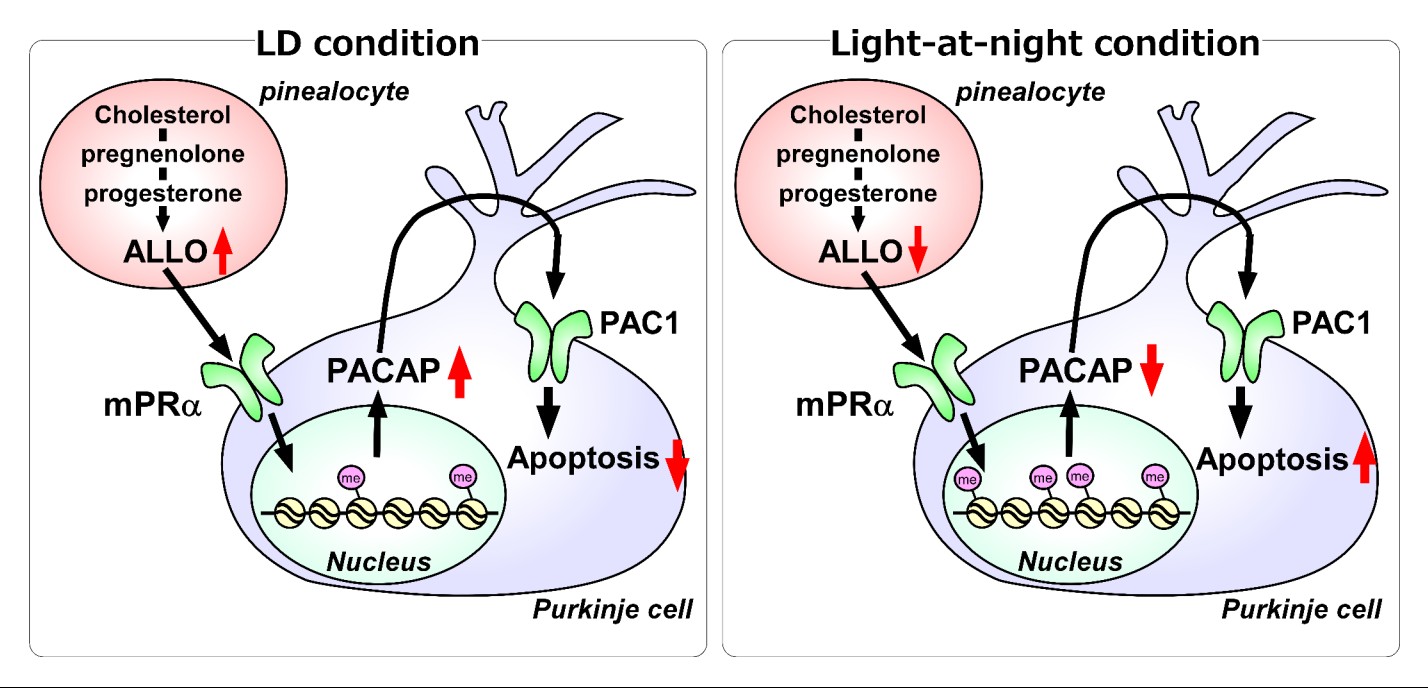

**Figure 11.** A schematic model of the effect of pineal ALLO on Purkinje cell survival during the first week posthatch under LD or light-at-night conditions. (Left panel) A schematic model of normal development of the cerebellum under LD during the first week posthatch (that is early posthatch life-stage). Pineal ALLO-induced the expression of PACAP, a neuroprotective factor, through mPRα mechanism in Purkinje cells. Then, PACAP inhibited the expression of active caspase-3 that may facilitate the apoptosis of Purkinje cells in the cerebellum. (Right panel) A schematic model of the abnormal development of the cerebellum under light-at-night conditions during the first week posthatch (that is early posthatch life-stage). Light-at-night conditions disrupted the diurnal rhythm in pineal ALLO synthesis. Decreased pineal ALLO synthesis induced H3K9me3 histone tail modifications of the *Adcyap1* gene promoter and then decreased the expression of PACAP in Purkinje cells. Following this, increased amounts of caspase-3 facilitated the apoptosis of Purkinje cells in the cerebellum.
DOI: https://doi.org/10.7554/eLife.45306.029

the perinatal periods has been shown to cause significant impairments to the structural development and organization of the brain (*Bernal, 2007*; *Koibuchi and Chin, 2000*; *Pasquini et al., 1967*). PACAP has important roles in the regulation of the cell cycle by enhancing cellular survival and by enhancing or inhibiting cellular proliferation and differentiation in the developing cerebellum (*Falluel-Morel et al., 2005*; *Vaudry et al., 2003*). Cytochrome P450 aromatase knock-out (ArKO) mice exhibited decreased Purkinje dendritic growth, spinogenesis, and synaptogenesis, and these changes were rescued by injection of estradiol into the cerebellum (*Haraguchi et al., 2012a*; *Sasahara et al., 2007*). Thus, a delicate balance is necessary for the proper development of the brain. It seems reasonable that the diurnal rhythm of pineal ALLO synthesis and its release under LD conditions are essential for the normal development of the cerebellum during early life (*Figure 11*).

In the developing cerebellum, PACAP was expressed not only by Purkinje cells but also by the cells of the molecular layer. However, mPRα, a receptor for ALLO, was only expressed in Purkinje cells. This result suggested that pineal ALLO mainly affects the Purkinje cells through mPRα in the developing cerebellum (*Figure 11*). PACAP secreted from Purkinje cells affects the PAC1-expressing Purkinje and granule cells. In Purkinje cells, PACAP mediates the neuroprotective effects of ALLO in the developing cerebellum of chicks (*Figure 11*). However, the effect of PACAP on granule cells is unclear in the developing cerebellum of chicks, although the neuroprotective effects of PACAP on cerebellar granule cells have been reported in mammals (*Vaudry et al., 2000*; *Vaudry et al., 2003*).

Several lines of evidence indicate that ALLO has neuroprotective effects in the developing and mature brain and in neurodegenerative diseases, including Parkinson's, Alzheimer's, and Huntington's disease (*Griffin et al., 2004*; *Irwin and Brinton, 2014*; *Melcangi and Panzica, 2014*). Numerous studies have found that ALLO exerts its various physiological functions through the $GABA_A$ receptors in the brain (*Belelli and Lambert, 2005*; *Guennoun et al., 2015*); however, the results of

the present study show that GABA$_A$ receptors did not mediate the neuroprotective effects of ALLO in developing Purkinje cells in chicks during early life. Other studies have also suggested that GABA$_A$ receptors do not contribute to the neuroprotective effects of ALLO. Recent investigations have suggested that ALLO binds to mPRs (*Pang et al., 2013*; *Schumacher et al., 2014*). Thus, we investigated whether mPRs can function as a receptor for ALLO to mediate the neuroprotective action of ALLO in the cerebellum. The results showed that ALLO had an important role in developing Purkinje cells through the mPRα mechanism in chicks during early life.

The neuroprotective mechanisms of ALLO are poorly understood, although numerous studies have found that ALLO prevents apoptotic neuronal cell death in the brain of vertebrates. In contrast to ALLO, the neuroprotective mechanisms of PACAP are well documented. The neurotrophic effects of PACAP are mediated through the cAMP/PKA signaling pathway and involve the ERK MAP kinase (*Vaudry et al., 2000*). Caspase-3 is also a key enzyme involved in the neuroprotective action of PACAP (*Vaudry et al., 2000*). We previously reported that the neuroprotective action of pineal ALLO was associated with a reduction in caspase-3 activity during the early stage of cerebellar development (*Haraguchi et al., 2012a*). In this study, we showed that PACAP expression showed clear diurnal changes in the developing cerebellum. Previously, PACAP levels in the brain have been shown to undergo daily variations (*Józsa et al., 2001*). Here, we showed that Px decreased the expression of PACAP in the developing cerebellum. In addition, previous studies have also reported that the expression levels of PACAP in the brain were affected by Px (*Somogyvári-Vigh et al., 2002*). These previous findings are in agreement with the present findings indicating that PACAP, a neuroprotective hormone, mediated the neuroprotective action of ALLO during early life.

Histone modifications store epigenetic information that mainly controls heritable states of gene expression. Environmental factors affect the developing brain through epigenetic mechanisms (*Bale, 2015*). In this study, we demonstrated that light-at-night and pineal ALLO changed the repressive mark (H3K9me3) of the *Adcyap1* gene promoter in the developing cerebellum. Our results indicated that inappropriate light conditions induced an abnormal epigenetic status in the developing cerebellum, which suggests that light conditions and pineal hormones have important roles in brain development via epigenetic regulation of hormone genes.

In conclusion, our results show that light-at-night-induced circadian disruption led to cerebellar Purkinje cell death through pineal ALLO-dependent mechanisms during early posthatch life (*Figure 11*). Thus, the results suggest that modern nighttime artificial light exposure also affects development of the human brain.

# Materials and methods

**Key resources table**

| Reagent type (species) or resource | Designation | Source or reference | Identifiers | Additional information |
|---|---|---|---|---|
| Cell line (*Cercopithecus aethiops*) | COS-7 | JCRB | Cat# JCRB9127 RRID:CVCL_0224 | |
| Recombinant DNA reagent | membrane progesterone receptor α (mPRα)-Histag | This study | | Materials and methods subsection 'Conformation of the specificity of mPRα antisera' |
| Antibody | Anti-Cleaved Caspase-3 (Asp175) , (Rabbit polyclonal) | Cell Signaling Technology | Cat# 9661, RRID:AB_2341188 | 1:300, IHC |
| Antibody | Anti-Calbindin D-28k, (Mouse monoclonal) | Swant Swiss antibodies | Cat# 300, RRID:AB_10000347 | 1:1000, IHC |
| Antibody | Anti-Pituitary adenylate cyclase-activating polypeptide (PACAP), (Chicken polyclonal) | *Nakamachi et al., 2018* | DOI: 10.1016/j.peptides.2018.03.006 | 1:100, IHC |

*Continued on next page*

*Continued*

| Reagent type (species) or resource | Designation | Source or reference | Identifiers | Additional information |
|---|---|---|---|---|
| Antibody | Anti-membrane progesterone receptor α (mPRα), (Rabbit polyclonal) | This study | | Materials and methods subsection 'Conformation of the specificity of mPRα antisera'; Against Goldfish mPRα aa 19–34 and 53–66; 1:100, IHC |
| Antibody | Anti-ADCYAP1R1 (PAC1), (Mouse monoclonal) | Santa Cruz Biotechnology | Cat# sc-100315, RRID:AB_1126992 | 1:100, IHC |
| Antibody | Anti-trimethyl-Histone H3 (Lys9), (Rabbit polyclonal) | Merck Millipore | Cat# 07–442, RRID:AB_310620 | 1:50, ChIP |
| Antibody | Anti-trimethyl-Histone H3 (Lys27), (Rabbit polyclonal) | Merck Millipore | Cat# 07–449, RRID:AB_310624 | 1:50, ChIP |
| Antibody | Anti-trimethyl-Histone H4 (Lys20), (Rabbit polyclonal) | Merck Millipore | Cat# 07–463, RRID:AB_310636 | 1:50, ChIP |
| Antibody | Anti-Mouse IgG (H+L), F(ab')₂ Fragment, Alexa Fluor 555 Conjugate | Cell Signaling Technology | Cat# 4409, RRID:AB_1904022 | 1:1000, IHC |
| Antibody | Anti-rabbit IgG (H+L), F(ab')₂ Fragment, Alexa Fluor 488 Conjugate | Cell Signaling Technology | Cat# 4412, RRID:AB_1904025 | 1:1000, IHC |
| Peptide, recombinant protein | Chicken PACAP | This study | | |
| Commercial assay kit | SimpleChIP plus sonication chromatin IP kit | Cell Signaling Technology | Cat# 56383 | |
| Chemical compound | Allopregnanolone | Cayman Chemical | 16930, CAS RN: 516-54-1 | |
| Chemical compound, drug | Fluorescein-labeled allopregnanolone | This study | CAS RN: 2294937-67-8 | Materials and methods subsection 'Fluorescein-labeled ALLO synthesis' |
| Software, algorithm | GraphPad Prism6 | GraphPad Software | RRID:SCR_002798 | |

## Animals

Domestic chickens (*Gallus gallus*) of both sexes at various ages were used in this study. Chicks were incubated under either a 12 hr light–12 hr dark (LD) cycle, constant light (LL), or a 12 hr light–12 hr dark cycle followed by exposure to light for 1 hr from ZT14 to ZT15 (light-at-night) with the light provided by white fluorescent lamps. The experimental protocols (2011-A090, 2012-A003, 2013-A010, 2014-A063, 2015-A012, 29M050, 30M047) were in accordance with the Guide for the Care and Use of Laboratory Animals of Waseda University or Showa University, Japan.

## Quantification of mRNAs

To measure the level of mRNA expressions, real-time PCR was performed using the StepOnePlus system (Applied Biosystems, Foster City, CA) as described previously (*Haraguchi et al., 2010*; *Haraguchi et al., 2012b*; *Haraguchi et al., 2012b*; *Haraguchi et al., 2015*; *Nozaki et al., 2018*). The sequence of each primer is shown in *Table 1*. *Gapdh* was used as the internal standard. The reaction mixture contained SYBR Green Real-Time PCR Mix (Toyobo, Osaka, Japan), 400 nM of forward and reverse primers, and 30 ng of cDNA in a final volume of 20 μL. PCR was run with a standard cycling program: 95℃ for 3 min; 40 cycles of 95℃, 15 s; 60℃, 15 s; and 72℃, 15 s. An external

**Table 1.** Oligonucleotide sequence of PCR.

| Target genes | Forward primer 5'- > 3' | Reverse primer 5'- > 3' |
| --- | --- | --- |
| Srd5a | AGAAAACCCGGGGGAAGTCAC | AGCGATGGCAAAACCAAACC |
| α1-subunit of GABA$_A$R | TCGTGGCAGTCTCCTTTGTC | CTCATGCCCACAAGTGTCCT |
| mPRα | TCTGCCCTGTGTGTCTTCAC | TTTGTCCCTCACCTTCCGTG |
| mPRβ | AGGGCCTTGTGGGAAAGATG | TGCCAGATTCAAAGCCCCAT |
| mPRγ | CGTGCGCTCGATGAGAAATG | TTTCATAACCCACCCCCAGC |
| PXR | CCCATAACCAAAGCCAAGCG | ATCATGTCCTTCCGCATCCC |
| PACAP | CACGCCGATGGGATCTTCA | GTGCAGGTATTTCCTTGCGG |
| BDNF | ACATCACTGGCGGACACTTT | CAGCATGACTCGGGATGTGT |
| NT-3 | ACCACCACCACTGTACCTCA | TCGGTGGCTCTTGTGTTCTG |
| 3β-HSD | CACTCTGCTGAACACCCCTT | GCTGGTGTACCTCTTTGCCT |
| P450arom | CGGGGCTGTGTAGGAAAGTT | TGTCTGTACTCTGCACCGTC |
| Dio2 | TGACCACCATTCACAAGCCA | CAACAGAAAGTCAGCCACGC |
| IGF-1 | ATGGATCCAGCAGTAGACGC | GCCTCCTCAGGTCACAACTC |
| *Adcyap1* gene promoter | CAGTTTCATGGTAAGGACCCG | ACGACCCACCGAGCG |

DOI: https://doi.org/10.7554/eLife.45306.030

standard curve was generated by a serial 10-fold dilution of cDNA obtained from the salmon brain, which had been purified, and its concentration was measured. To confirm the specificity of the amplification, the PCR products were subjected to melting curve analysis and gel electrophoresis. The results were normalized to the expression of *gapdh* using StepOnePlus 2.0 software (Applied Biosystems).

## Measurement of ALLO synthesis

To assess ALLO formation in the chick pineal gland, the conversion of progesterone, a substrate of ALLO, was measured biochemically using pineal gland homogenates. The biochemical analyses in this study were performed as described previously (*Doi et al., 2010*; *Haraguchi et al., 2010*; *Haraguchi et al., 2012a*; *Haraguchi et al., 2012b*; *Haraguchi et al., 2015*; *Nozaki et al., 2018*). In brief, 10 mg of pineal gland homogenates was incubated in phosphate-buffered saline (PBS) containing $10^6$ cpm [$^3$H] progesterone for 0 or 60 min at 37˚C. After incubation, steroids were extracted by ethyl acetate and subjected to high-performance liquid chromatography analysis using a reversed-phase column, LiChrospher 100 RP-18 (Kanto Kagaku, Tokyo, Japan). Tritiated ALLO was used as the standard to detect the elution positions. All tritiated steroids were purchased from PerkinElmer (PerkinElmer Japan, Yokohama, Japan).

## Measurement of ALLO concentration

ALLO concentration in the chick pineal gland was measured by liquid chromatography-electrospray ionization-tandem mass spectrometry (LC-ESI-MS/MS; QTRAP 4500 LC-MS/MS System; AB Sciex, Foster City, CA). In brief, 5 mg of pineal gland tissue was homogenized in 1 mL methanol/H$_2$O (75:25; vol/vol) on ice. The homogenate was loaded on Isolute SLE+ cartridges (Biotage, Uppsala, Sweden) for supported liquid–liquid extraction, and the steroid fractions were eluted with dichloromethane and subjected to LC-ESI-MS/MS analysis.

## Quantification of the number of Purkinje cells and the thickness of the molecular layer

To investigate whether ALLO and PACAP are involved in Purkinje cell survival in birds, light-at-night or Px chicks were injected daily with either ALLO or PACAP.

### Light-at-night chicks

Daily injection of ALLO (30 ng/5 µL sesame oil) into light-at-night chicks was performed into the pineal gland region of the brain from ZT11 to ZT12 once per day for 7 days during the P1–P7 period. Control treatment consisted of an equal volume of vehicle-only.

### Px chicks

Px and sham operations on P1 chicks were performed under isoflurane anesthesia as described previously (*Haraguchi et al., 2012a*). After surgery, either ALLO (30 ng/5 µL) dissolved in sesame oil or PACAP ($10^{-8}$ M/5 µL) dissolved in physiological saline was injected into the pineal gland region of the brain in chicks from ZT11 to ZT12 once per day for 7 days during the P1–P7 period. Control animals were treated with an equal volume of vehicle-only.

The injection site was determined by visually inspecting the brain of a chick injected with 5 µL of 0.15% methylene blue dissolved in saline. After decapitation under deep anesthesia, the cerebella of the control, Px, or ALLO- or PACAP-treated chicks at P21 were dissected. Cerebellar sections from 4% paraformaldehyde-fixed cerebella were processed and stained with anti-calbindin D-28k (300; Swant Swiss antibodies, Bellinzona, Switzerland) as described previously (*Haraguchi et al., 2012a*). The number of calbindin-immunoreactive cell bodies of Purkinje cells was counted in each lobule at P21 or P70. All analyses were performed in the cerebellar vermis area. The number of Purkinje cells in the vermis area was calculated from 10 sections (20 µm thickness, every three sections in the vermis area) per animal. The length of the molecular layer in the parasagittal section was evaluated as the thickness of the molecular layer as described previously (*Haraguchi et al., 2012a*).

## Assessment of neuronal cell death

Parasagittal cerebellar sections of chicks at P7 were examined by immunostaining with an antibody against cleaved caspase-3 (Asp175; #9661; Cell Signaling Technology, Danvers, MA) to detect apoptotic cells immunoreactive to the active form of caspase-3 as described previously (*Haraguchi et al., 2012a*). The number of cleaved caspase-3-positive Purkinje cells in the vermis area was calculated from 10 sections (20 µm thickness, every three sections in the vermis area) per animal.

## RT-PCR analysis

Total RNA was extracted from the chick cerebellum using Sepazol-RNA I Super (Nacalai Tesque, Kyoto, Japan) and reverse transcribed as described previously (*Haraguchi et al., 2010*; *Haraguchi et al., 2012a*; *Haraguchi et al., 2012b*; *Haraguchi et al., 2015*; *Nozaki et al., 2018*). All PCR amplifications (for mPRα, mPRβ, mPRγ, PXR, and α1-subunit of the GABA$_A$ receptor) were performed in a reaction mixture containing *Ex Taq* polymerase (Takara, Shiga, Japan). Forward and reverse primers (*Table 1*) were designed according to the nucleotide sequence of chicken mRNAs. The following PCR conditions were used for the thermal cycler: 1 cycle of 1 min at 94℃, 30 cycles of 30 s at 94℃, 30 s at 60℃, 30 s at 72℃, and finally, 1 cycle of 10 min at 72℃. The identities of the PCR products were confirmed by sequencing.

## *In ovo* transfection by electroporation

For the intracerebellar injection of siRNAs, *in ovo* transfections were performed using chick embryos. The sequence of each siRNA is shown in *Table 2*. In vivo transfection by electroporation for chick embryos was performed as described previously (*Katahira and Nakamura, 2003*). Each siRNA reagent was injected into the cerebellar cortex of lobule IV. Subsequently, a pair of stainless steel

**Table 2.** Oligonucleotide sequence of siRNA.

| Target genes | Sequence-sense | Sequence-anti-sense |
| --- | --- | --- |
| mPRα | CGGAGCUGGGCUGGUUUCUUCCCA | UGGGAAGAAACCAGCCCAGCUCCG |
| mPRβ | GAGGAGGAUGCUGCUUGGUACCAU | AUGGUNCCAAGCAGCAUCCUCCUC |
| mPRγ | CCGACAGAGUUUGGCUGCUGCGAU | AUCGCAGCAGCCAAACUCUGUCGG |

DOI: https://doi.org/10.7554/eLife.45306.031

electrodes was inserted into the neural tube and then transfected with an electroporator (NEPA21 Electroporator; NEPA Gene, Chiba, Japan).

## Conformation of the specificity of mPRα antisera

To confirm that the mPRα antiserum recognized the appropriate antigen, western blot analysis was performed on extracts of COS-7 cells transfected with chicken mPRα cDNA as described previously (*Haraguchi et al., 2010*; *Haraguchi et al., 2012a*; *Haraguchi et al., 2012b*; *Haraguchi et al., 2015*; *Nozaki et al., 2018*). COS-7 cells (Cat# JCRB9127, RRID:CVCL_0224) were grown in DMEM with 10% fetal bovine serum. Cell line authentication and mycoplasma-free certification was performed by JCRB. The extract of COS-7 cells transfected with chicken mPRα cDNA was separated on a 12.5% SDS-polyacrylamide gel under reducing conditions and transferred to PVDF membranes (GE Healthcare, Madison, WI). The membrane was incubated with anti-goldfish mPRα antiserum at 4°C overnight and then for 1 hr with goat-anti-rabbit IgG-horseradish-peroxidase conjugate diluted 1:1000. An intense immunoreaction band was detected using an ImmunoStar LD western blotting detection system (Wako Pure Chemicals, Osaka, Japan).

## IHC staining

IHC localizations of mPRα, PACAP, or PAC1 were performed as described previously (*Haraguchi et al., 2010*; *Haraguchi et al., 2012a*; *Haraguchi et al., 2012b*; *Haraguchi et al., 2015*; *Nozaki et al., 2018*). The cerebella were fixed in 4% (vol/vol) paraformaldehyde solution overnight and then soaked in a refrigerated 30% (vol/vol) sucrose solution in 0.1 M PB. Cerebella were frozen in OCT compound (Miles, Elkhart, IN) and then sectioned transversely at 10 μm thickness on a cryostat at −20°C. After blocking nonspecific binding with Protein Block, serum-free (Agilent Technologies, Palo Alto, CA), the sections were immersed overnight at 4°C in either a 1:100 dilution of rabbit anti-goldfish mPRα antiserum, a 1:1000 dilution of chicken anti-PACAP antibody (*Nakamachi et al., 2018*), or a 1:100 dilution of mouse anti-PAC1 antibody (sc-100315; Santa Cruz Biotechnology, Santa Cruz, CA). The sections were then incubated for 60 min with Alexa Fluor 555-labeled second antibody (Cell Signaling Technology) at a dilution of 1:1000. After washing, the sections were mounted with mounting medium and visualized using a fluorescence microscope.

## Binding assay

To investigate the binding of ALLO to chicken mPRα, a radioreceptor assay was performed as described previously (*Ito et al., 2011*). The membrane fractions extracted from COS-7 cells transfected with chicken mPRα cDNA were incubated for 30 min at 4°C with 1–25 nM [$^3$H] ALLO in the presence or absence of cold ALLO competitor. At the end of the incubation, the samples were centrifuged at 10,000 $g$ for 10 min at 4°C. The supernatant was aspirated out, and the radioactivity of the pellets was counted using a liquid scintillation counter.

In addition, the cellular localization of fluorescein-labeled ALLO in COS-7 cells transfected with chicken mPRα cDNA was analyzed as described previously (*Haraguchi et al., 2010*; *Haraguchi et al., 2012a*; *Haraguchi et al., 2012b*; *Haraguchi et al., 2015*; *Nozaki et al., 2018*). The COS-7 cells were treated with fluorescein-labeled ALLO in the presence or absence of cold ALLO for 1 hr at 4°C. At the end of the incubation, the cells were fixed for 1 hr in 4% paraformaldehyde and then washed 3 times for 5 min in PBS. After washing, the cells were mounted with mounting medium and visualized using a confocal microscope.

## Chromatin immunoprecipitation and quantitative PCR (ChIP-qPCR)

Chromatin immunoprecipitation (ChIP) was performed using the SimpleChIP plus sonication chromatin IP kit (Cell Signaling Technology) according to the manufacturer's protocol. Briefly, the Purkinje cell layer, including Purkinje cells, was laser-microdissected (LS-AMD; Leica Microsystems, Bensheim, Germany). The crosslinked cells of the Purkinje cell layer were sonicated to shear the chromatin to 200–1,000 bp. Each IP was performed using rabbit polyclonal anti-trimethyl-histone H3 (Lys9) antibody (07–442; Merck Millipore, Schwalbach, Germany), rabbit polyclonal anti-trimethyl-histone H3 (Lys27) antibody (07–449; Merck Millipore), or rabbit polyclonal anti-trimethyl-histone H4 (Lys20) antibody (07–463; Merck Millipore). qPCR analyses were performed using the StepOnePlus 2.0 software (Applied Biosystems) as described above. The sequence of each primer is shown in *Table 1*.

## Statistical analysis

GraphPad Prism (GraphPad) and Estimation statistics (http://www.estimationstats.com) were used for statistical analysis. Student's two-tailed $t$-test, or one-way ANOVA followed by post hoc analysis with Tukey's test for multiple comparisons test were performed. P values of the relevant post hoc multiple comparisons are shown in the figures. The statistical significance cutoff was set at p<0.05.

## Fluorescein-labeled ALLO synthesis

$^1$H NMR and $^{13}$C NMR spectra were recorded with CDCl$_3$ as the solvent using tetramethylsilane as an internal standard on JEOL AL-400 spectrometers (JEOL, Akishima, Japan). Multiplicities are indicated as br (broadened), s (singlet), d (doublet), t (triplet), q (quartet), and m (multiplet). High-resolution mass spectrometry spectra were recorded on JMS-SX102A (JEOL). All of the isolated materials were shown to be pure by NMR (free of obvious impurities) and thin-layer chromatography (homogeneous material).

First, we prepared ALLO (**2**) as described previously (*Comin et al., 2004*). Then, we followed Schemes 1–6 to synthesize fluorescein-labeled ALLO (**12**).

## 3-*p*-Toluenesulfonyloxy-ALLO (3)

To the solution of ALLO **2** (1.00 g, 3.14 mmol) in pyridine (10 mL) was added *p*-toluenesulfonyl chloride (1.20 g, 6.28 mmol) at room temperature, and the mixture was stirred for 12 hr. The aqueous phase was extracted with ethyl acetate. The combined organic layers were dried (MgSO$_4$), and the solvent was evaporated. The residue was purified by column chromatography (silica gel) using hexane–EtOAc (2:1) as eluent to afford 1.19 g (80% yield) of tosylate **3**.

$^1$H NMR (400 MHz, CDCl$_3$) δ: 7.68–7.75 (d, $J$ = 8.2 Hz, 2H, Ar-H), 7.21–7.29 (d, $J$ = 8.6 Hz, 2H, Ar-H), 4.30–4.40 (m, 1H, 3 C-H), 2.39–2.47 (m, 1H, 17 C-H), 2.37 (s, 3H, Ts-Me), 2.03 (s, 3H, 19C-Me), 0.70 (s, 3H, 10C-Me), 0.51 (s, 3H, 13C-Me).

## 3β-acetoxy ALLO (4)

To the solution of tosylate **3** (47 mg, 0.0995 mmol) in 2-butanone (2 mL) was added tetra-*n*-butylammonium acetate (105 mg, 0.34 mmol), and the mixture was stirred at reflux for 20 hr. The reaction mixture was quenched with water and extracted with ether. The extracts were washed with brine, dried over MgSO$_4$, filtered, and concentrated *in vacuo*. The residue was purified by flash column chromatography on silica gel (hexane:ethyl acetate, 2:1) to give acetate **4** (20 mg, 56%).

$^1$H NMR (400 MHz, CDCl$_3$) δ 5.00 (s, 1H, 3 C-H), 2.48–2.54 (t, $J$ = 9.1 Hz, 1H, 17 C-H), 2.10 (s, 3H, Ac-Me), 2.05 (s, 3H, 19C-Me), 0.78 (s, 3H, 10C-Me), 0.60 (s, 3H, 13C-Me).

## Alyl 3-((3*R*,5*S*,8*S*,10*S*,13*S*,14*S*,17*S*)−3-acetoxy-10,13-dimethylhexadecahydro-1*H*- cyclopenta[*a*]phenanthren-17-yl)−3-oxopropanoate (5)

To a solution of **4** (6.40 g, 11.82 mmol) in THF (14.6 mL) was added LHMDS (1.3 M in THF, 22.75 mL) at −78°C. After 1 hr, allyl chloroformate (1.38 mL, 13.0 mmol) was added, and the mixture was stirred at −78°C for 2 hr. To the reaction was added saturated aqueous NH$_4$Cl, and the organic layer was partitioned. The aqueous phase was extracted with ether, and the combined organic extracts were dried with MgSO$_4$, filtered, and concentrated *in vacuo*. The residue was purified by flash column chromatography on silica gel (hexane/ethyl acetate = 95:5) to give **5** (3.78 g, 6.054 mmol, 51%) as a white solid.

$^1$H NMR (400 MHz, CDCl$_3$) δ 8.02 (d, $J$ = 6.3 Hz, 4H, Ar-H), 7.57 (dd, $J$ = 6.3 Hz, 1H, Ar-H), 7.54 (dd, $J$ = 6.3 Hz, 1H, Ar-H), 7.46 (dd, $J$ = 6.3 Hz, 2H, Ar-H), 7.42 (dd, $J$ = 6.3 Hz, 2H, Ar-H), 5.92 (m, 1H), 5.78 (d, $J$ = 4.8 Hz, 1H, 6 C-H), 5.33 (dd, $J$ = 17, 1.5 Hz, 1H), 5.28 (br, 1H, 7 C-H), 5.24 (dd, $J$ = 10, 1.5 Hz, 1H), 4.90 (m, 1H, 3 C-H), 4.63 (dt, $J$ = 5.8, 1.2 Hz, 2H), 3.47 (s, 1H, 21C-CH), 1.12 (s, 3H, 19C-CH$_3$), 0.68 (s, 3H, 18C-CH$_3$).

## (3*R*,5*S*,8*S*,10*S*,13*S*,14*S*,17*S*)−10,13-Dimethyl-17-(pent-4-enoyl)hexadecahydro-1*H*-cyclopenta[*a*]phenanthren-3-yl acetate (6)

To a solution of **5** (3.78 g, 6.05 mmol) in dioxane (60 mL) were added Pd$_2$(dba)$_3$•CHCl$_3$ (0.313 g, 0.303 mmol) and DPPE (241.2 mg, 0.605 mmol), and the mixture was stirred at room temperature

for 4 hr. The reaction mixture was filtered through a pad of Celite, and the filtrates were concentrated *in vacuo*. The residue was purified by flash column chromatography on silica gel (hexane/ethyl acetate = 97:3) to give **6** (2.92 g, 5.03 mmol, 83%) as a white solid.

[1]H NMR (400 MHz, CDCl$_3$) $\delta$ 8.02 (d, $J$ = 6.3 Hz, 4H, Ar-H), 7.57 (dd, $J$ = 6.3 Hz, 1H, Ar-H), 7.54 (dd, $J$ = 6.3 Hz, 1H, Ar-H), 7.46 (dd, $J$ = 6.3 Hz, 2H, Ar-H), 7.42 (dd, $J$ = 6.3 Hz, 2H, Ar-H), 5.81 (m, 1H, 23 C-H), 5.77 (d, $J$ = 3.6 Hz, 1H, 6 C-H), 5.27 (t, $J$ = 4.1 Hz, 1H, 7 C-H), 5.02 (ddd, $J$ = 17, 3.4, 1.7 Hz, 1H, 24 C-H), 4.96 (ddd, $J$ = 10, 2.9, 1.2 Hz, 1H, 24 C-H), 4.90 (m, 1H, 3 C-H), 1.11 (s, 3H, 19C-CH$_3$), 0.67 (s, 3H, 18C-CH$_3$).

### 6-Amino-3-oxo-3*H*-spiro[isobenzofuran-1,9′-xanthene]−3′,6′-diyl bis(2,2-dimethylpropanoate) (8)

To a stirred solution of 5-aminofluorescein **7** (0.10 g, 0.288 mmol) in DMF (3.0 mL) were added Cs$_2$CO$_3$ (0.28 g, 0.864 mmol) and pivalic anhydride (0.13 mL, 1.90 mmol), and the mixture was stirred at room temperature for 1 hr. The reaction mixture was filtered through a pad of Celite, and the filtrates (DMF layer) were extracted with ether. The extracts were concentrated *in vacuo*, and the residue was purified by flash column chromatography on silica gel (CH$_2$Cl$_2$:water, 4:1) to give **8** (87.4 mg, 0.17 mmol, 59%) as a white solid.

[1]H NMR (400 MHz, CDCl$_3$) $\delta$ 7.76 (d, $J$ = 8.3 Hz, 1H, Ar-H), 7.01 (d, $J$ = 2.2 Hz, 2H, Ar-H), 6.96 (s, 1H, Ar-H), 6.94 (s, 1H, Ar-H), 6.78 (dd, $J$ = 8.5, 2.2 Hz, 2H, Ar-H), 6.76 (dd, $J$ = 8.3, 1.9 Hz, 1H, Ar-H), 4.21 (brs, 2H, NH$_2$), 1.36 (s, 18H, C(CH$_3$)$_3$).

### 3-Oxo-6-(undec-10-enamido)−3*H*-spiro[isobenzofuran-1,9′-xanthene]−3′,6′-diyl bis(2,2-dimethylpropanoate) (9)

To a stirred solution of **8** (0.10 g, 0.288 mmol) in CH$_2$Cl$_2$ (1.0 mL) were added DMAP (37 mg, 0.305 mmol), DIC (0.048 mL, 0.305 mmol), and undecenoic acid (0.0513 mL, 0.254 mmol), and the mixture was stirred at room temperature for 50 hr. To the reaction mixture was added saturated aqueous NaHCO$_3$, and the organic layer was partitioned. The aqueous layer was extracted with CH$_2$Cl$_2$, and the combined organic extracts were dried with MgSO$_4$, filtered, and concentrated *in vacuo*. The residue was purified by flash column chromatography on silica gel (hexane:ethyl acetate, 4:1) to give **9** (78.86 mg, 0.12 mmol, 68%) as a white solid.

[1]H NMR (400 MHz, CDCl$_3$) $\delta$ 7.92 (d, $J$ = 8.6 Hz, 1H, Ar-H), 7.85 (dd, $J$ = 8.6, 1.6 Hz, 1H, Ar-H), 7.58 (br, 1H, NH), 7.21 (s, 1H, Ar-H), 7.00 (d, $J$ = 2.2 Hz, 2H, Ar-H), 6.85 (d, $J$ = 8.6 Hz, 2H, Ar-H), 6.76 (dd, $J$ = 8.6, 2.2 Hz, 2H, Ar-H), 5.78 (m, 1H), 4.96 (dd, $J$ = 17, 1.6 Hz, 1H), 4.91(dd, $J$ = 12, 1.6 Hz, 1H), 2.27 (t, $J$ = 7.5 Hz, 2H, COCH$_2$), 2.00 (dd, $J$ = 7.3, 6.7 Hz, 2H), 1.35 (s, 18H, C(CH$_3$)$_3$).

### (3*R*,5*S*,8*S*,10*S*,13*S*,14*S*,17*S*)−10,13-Dimethyl-17-((E)−14-oxo-14-(3-oxo-3′,6′-bis(pivaloyloxy)−3H-spiro[isobenzofuran-1,9′-xanthene]−6-ylamino)tetradec-4-enoyl)- (pent-4-enoyl)hexadecahydro-1*H*-cyclopenta[*a*]phenanthren-3-yl acetate (10)

To a stirred solution of **7** (241.9 mg, 0.604 mmol) and **9** (412.02 mg, 0.604 mmol) in CH$_2$Cl$_2$ (3.5 mL) was added Hoveyda Grubbs' 2$^{nd}$ catalyst (0.0031 g, 0.0036 mmol), and the resulting mixture was heated under reflux. After 26 hr, the mixture was filtered through a pad of Celite, and the organic solvent was evaporated. The crude product was purified by flash column chromatography (10% to 25% ethyl acetate/hexane) to afford **10** (30.0 mg, 0.028 mmol, 5%).

[1]H NMR (400 MHz, CDCl$_3$) $\delta$ 7.90 (d, $J$ = 8.2 Hz, 1H, Ar-H), 7.86 (dd, $J$ = 8.7, 1.4 Hz, 1H, Ar-H), 7.23 (br, 1H, Ar-H), 7.00 (d, $J$ = 1.8 Hz, 2H, Ar-H), 6.85 (d, $J$ = 8.7 Hz, 2H, Ar-H), 6.76 (dd, $J$ = 8.7, 2.3 Hz, 2H, Ar-H), 5.39–5.31 (m, 2H), 5.00 (br, 1H), 4.21 (t, $J$ = 5.9 Hz, 1H), 2.48 (t, $J$ = 8.7 Hz, 1H), 2.04 (s, 3H), 1.34 (s, 18H), 0.78 (s, 3H), 0.56 (s, 3H).

### (3*R*,5*S*,8*S*,10*S*,13*S*,14*S*,17*S*)−10,13-Dimethyl-17–14-oxo-14-(3-oxo-3′,6′-bis(pivaloyloxy)−3H- spiro[isobenzofuran-1,9′-xanthene]−6-ylamino)tetradecanoyl)-(pent-4-enoyl)hexadecahydro-1*H*-cyclopenta[*a*]phenanthren-3-yl acetate (11)

Compound **10** (30.0 mg, 0.028 mmol) in ethyl acetate (0.28 mL) was hydrogenated over 10 wt% Pd on activated carbon (3.0 mg). After 3 hr, the reaction mixture was diluted with ethyl acetate and

filtered to remove the catalyst. The filtrate was concentrated, and the crude product **11** was used in the next step without further purification.

$^1$H NMR (400 MHz, CDCl$_3$) $\delta$ 7.92 (d, $J$ = 9.1 Hz, 1H, Ar-H), 7.86 (dd, $J$ = 8.6, 1.8 Hz, 1H, Ar-H), 7.69 (br, 1H, Ar-H), 7.01 (d, $J$ = 2.2 Hz, 2H, Ar-H), 6.86 (s, Ar-H), 6.86 (s, Ar-H), 6.78 (d, $J$ = 2.2 Hz, 1H, Ar-H), 6.76 (d, $J$ = 2.2 Hz, 1H, Ar-H), 5.01 (br, 1H), 2.45 (t, $J$ = 9.2 Hz, 1H), 2.37–2.24 (m, 4H), 2.18–2.11 (m, 1H), 2.05 (s, 3H), 1.99–1.92(m, 1H), 1.35 (s, 18H), 0.79 (s, 3H), 0.57 (s, 3H).

### Fluorescein-labeled ALLO (12)

To a stirred solution of **11** in MeOH (1.8 mL) was added K$_2$CO$_3$ (18.5 mg, 0.018 mmol), and the resulting mixture was heated under reflux for 16 hr. The reaction was quenched by the addition of saturated aqueous NH$_4$Cl, and the organic solvent was evaporated. The residue was extracted with ethyl acetate, and the combined organic extracts were dried with MgSO$_4$ and concentrated in vacuo. The crude product was purified by flash column chromatography (ethyl acetate in hexane) to afford **12** (11.1 mg, 0.013 mmol, 73%).

$^1$H NMR (400 MHz, CD$_3$OD) $\delta$ 7.92 (d, $J$ = 8.7 Hz, 1H, Ar-H), 7.76 (dd, $J$ = 8.2, 1.4 Hz, 1H, Ar-H), 7.57 (d, $J$ = 1.8, 1H, Ar-H), 6.68–6.63 (m, 4H, Ar-H), 6.56 (d, $J$ = 2.8 Hz, 1H, Ar-H), 6.53 (d, $J$ = 2.8 Hz, 1H, Ar-H), 4.54 (br, 2H), 3.92 (m, 1H), 3.34 (s, 3H), 2.60 (t, $J$ = 8.7 Hz, 1H), 0.78 (s, 3H), 0.56 (s, 3H).

## Acknowledgements

This work was supported in part by Japan Society for the Promotion of Science (JSPS) Grants-in-Aid for Scientific Research (15K18571 to SH and 22227002 to KT). This work was also supported by the Ichiro Kanehara Foundation, the Kao Research Council, the Naito Foundation, the Narishige Zoological Science Foundation, the Yamaguchi Endocrine Research Foundation, the Suntory Foundation for Life Sciences, and the Takeda Science Foundation (to SH). The authors would like to thank Enago (www.enago.jp) for the English language review.

## Additional information

### Funding

| Funder | Grant reference number | Author |
|---|---|---|
| Japan Society for the Promotion of Science | 15K18571 | Shogo Haraguchi |
| Takeda Science Foundation | | Shogo Haraguchi |
| Ichiro Kanehara Foundation for the Promotion of Medical Sciences and Medical Care | | Shogo Haraguchi |
| Kao Corporation | | Shogo Haraguchi |
| Naito Foundation | | Shogo Haraguchi |
| Narishige Zoological Science Foundation | | Shogo Haraguchi |
| Yamaguchi Endocrine Research Foundation | | Shogo Haraguchi |
| Suntory Foundation | | Shogo Haraguchi |
| Japan Society for the Promotion of Science | 22227002 | Kazuyoshi Tsutsui |

The funders had no role in study design, data collection and interpretation, or the decision to submit the work for publication.

### Author contributions

Shogo Haraguchi, Conceptualization, Data curation, Formal analysis, Supervision, Funding acquisition, Validation, Investigation, Writing—original draft, Project administration; Masaki Kamata,

Takuma Tokita, Data curation, Formal analysis, Investigation; Kei-ichiro Tashiro, Miku Sato, Mitsuki Nozaki, Data curation, Formal analysis, Investigation, Methodology; Mayumi Okamoto-Katsuyama, Resources, Data curation, Writing—review and editing; Isao Shimizu, Resources, Data curation, Formal analysis; Guofeng Han, Resources, Data curation, Investigation, Writing—review and editing; Vishwajit Sur Chowdhury, Resources, Data curation, Formal analysis, Investigation, Writing—review and editing; Xiao-Feng Lei, Formal analysis, Investigation, Writing—review and editing; Takuro Miyazaki, Joo-ri Kim-Kaneyama, Data curation, Formal analysis, Investigation, Writing—review and editing; Tomoya Nakamachi, Resources, Writing—review and editing; Kouhei Matsuda, Toshinobu Tokumoto, Resources, Investigation, Writing—review and editing; Hirokazu Ohtaki, Data curation, Software, Investigation, Visualization; Tetsuya Tachibana, Resources, Data curation, Formal analysis, Investigation, Methodology, Writing—review and editing; Akira Miyazaki, Investigation, Writing—review and editing; Kazuyoshi Tsutsui, Project administration, Writing—review and editing

## Author ORCIDs
Shogo Haraguchi (iD) https://orcid.org/0000-0002-8731-3311
Kouhei Matsuda (iD) http://orcid.org/0000-0002-8253-5230

## Ethics
Animal experimentation: The experimental protocols (2011-A090, 2012-A003, 2013-A010, 2014-A063, 2015-A012, 29M050, 30M047) were in accordance with the Guide for the Care and Use of Laboratory Animals of Waseda University or Showa University, Japan.

## Decision letter and Author response
Decision letter https://doi.org/10.7554/eLife.45306.036
Author response https://doi.org/10.7554/eLife.45306.037

# Additional files

## Supplementary files
• Transparent reporting form
DOI: https://doi.org/10.7554/eLife.45306.032

## Data availability
All data generated or analysed during this study are included in the manuscript and supporting files. Source data files for Figures 1-10 have been deposited to Dryad (https://doi.org/10.5061/dryad.k6g8b53).

The following dataset was generated:

| Author(s) | Year | Dataset title | Dataset URL | Database and Identifier |
|---|---|---|---|---|
| Haracguhi S, Kamata M, Tokita T, Tashiro K, Sato M, Nozaki M, Okamoto-Katsuyama M, Shimizu I, Han G, Chowdhury VS, Lei X, Miyazaki T, Kim-Kaneyama J, Nakamachi T, Matsuda K, Ohtaki H, Tokumoto T, Tachibana T, Miyazaki A, Tsutsui K | 2019 | Data from: Light-at-night exposure affects brain development through pineal allopregnanolone-dependent mechanisms | https://dx.doi.org/10.5061/dryad.k6g8b53 | Dryad Digital Repository, 10.5061/dryad.k6g8b53 |

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
