## [Decision Letter]

Thank you for submitting your article "Light affects cerebellar development through pineal allopregnanolone-dependent mechanisms during early posthatch life" for consideration by *eLife*. Your article has been reviewed by three peer reviewers, one of whom is a member of our Board of Reviewing Editors, and the evaluation has been overseen by K VijayRaghavan as the Senior Editor. The following individual involved in review of your submission has agreed to reveal their identity: Alanna J Watt. The reviewers have discussed the reviews with one another and the Reviewing Editor has drafted this decision to help you prepare a revised submission.

The study by Haraguchi et al. represents an important and interesting contribution to our understanding of the role of circadian rhythm in brain development. This work builds upon the authors' previous findings that pineal gland-derived ALLO hormone modulated Purkinje cell survival in the developing cerebellum of quail chicks (Haraguchi et al., 2012). Here, the authors use a light-at-night (LAN) paradigm combined with RT-PCR, competition assays, knock-downs with siRNA and IHC, to show that 1 hour of light at night reduces the expression of the pineal-derived hormone ALLO. They then go on to show downstream mechanisms for the neuroprotective action of ALLO: they identify the mPRalpha receptor through which ALLO appears to act, and show that this alters the expression of the neuroprotective hormone PACAP. Finally, they demonstrate that although PACAP can alter Purkinje cell survival during later development, its expression in the cerebellum decreases during development because of changes in histone modification levels at the adcyap1 gene promoter.

Though this study is interesting and thorough, the reviewers made several critical suggestions, which need to be satisfactorily addressed before the manuscript can be accepted for publication at *eLife*.

Essential revisions:

1) The extent of Purkinje neuron loss due to light-at-night in the current study seems to be small (Figure 1H). As it stands, it looks like there is a clear effect on pineal ALLO synthesis, but only a marginal effect on Purkinje cell survival. Small changes are not necessarily unimportant as even a decrease of 10% of cerebellar Purkinje cells could still have a big impact on cerebellar function. This raises two questions: one, if there are lobule-specific differences in Purkinje cell loss as seen in Haraguchi et al., 2012 with pinealectomy, and two if these small changes persist into adulthood and lead to cerebellar deficits. To delineate if there are lobule-specific differences, authors should provide lobule-specific counts of PNs after light-at-night. Such lobule-specific differences could potentially be quite interesting in terms of light regulation of Purkinje neuron survival. To see if these changes persist into adulthood, authors should look at counts at fully adult stages as well.

2) Data presentation: Throughout the manuscript data are presented as bar graphs showing mean+/- sem. It is now standard practice to show the scatter of the data with a suitable measure of central tendency depending on the shape of the distribution. Further, data in Figure 1G, H (and elsewhere in the paper, e.g. Figure 3B-E) are represented as relative to controls. It would be informative to have an indication of the actual number of cells. If necessary, this can be normalized to the total number of cells (or total fluorescence of a nuclear indicator) in each section.

3) The statistical analysis of the data is insufficient. It is not clear if the data are normal and normality tests were done, but a one-way ANOVA has been used. The authors should not take p<0.05 to indicate that their hypothesis is true. The statement by APA (Wasserstein, R. L. & Lazar, N. A. The ASA's Statement on p-Values: Context, Process, and Purpose. The American Statistician 70, 129-133 (2016)) should be consulted as to what the p-value can and cannot be used for. It is suggested that the authors use Gardner-Altman estimation plots (e.g. see estimationstats.com), and provide an indication of the size of the effect of the different treatments.

4) The authors should include example representative images for caspase-3 positive cells and Purkinje cells in Figure 2, so that the reader can evaluate these findings themselves.

5) Why were male chicks used exclusively? In previous studies by these authors, results were shown to be similar in both male and female chicks. These results will be stronger if they are confirmed in females. At minimum, a justification for the exclusive use of male animals should be provided.

6) Why has this particular light-at-night paradigm been used? It would be useful if the paradigm were described in the Materials and methods section (it is currently not mentioned there at all) and a brief discussion of the generalizability of this light-at-night paradigm could be included in the Discussion.

7) At what time was the ALLO injection given? The Materials and methods section mentions that vehicle only injections were done but it is not clear if the 'control' is vehicle-only or uninjected animals. Both conditions must be shown, with raw cell counts.

8) The authors go into some detail delineating the timeline of the action of light at night during development. However, it is hard to understand how this is believed to correspond to human (or mammalian) development, so it would be useful to discuss this more directly in the Discussion.

9) The second and third sentences of the Introductory paragraph could use some rewriting as they are only loosely connected to the rest of the text. "In particular, light conditions affect development during early life; therefore, it is not surprising that life evolved on this planet") is not logically sound.

[Editors' note: further revisions were requested prior to acceptance, as described below.]

Thank you for submitting your article "Light-at-night exposure affects brain development through pineal allopregnanolone-dependent mechanisms" for consideration by *eLife*. Your article has been reviewed by three peer reviewers, one of whom is a member of our Board of Reviewing Editors, and the evaluation has been overseen by K VijayRaghavan as the Senior Editor. The following individual involved in review of your submission has agreed to reveal their identity: Alanna J Watt (Reviewer #1).

The reviewers have discussed the reviews with one another and the Reviewing Editor has drafted this decision to help you prepare a revised submission.

The reviewers were in general happy with the revised manuscript and felt that the revisions made it much stronger. However, there are a few points which need to be rectified before the manuscript can be accepted. Please see these below in the comments section and address them at the earliest.

Reviewer #1:

The authors have submitted a revised manuscript of their work on light-at-night exposure affecting brain development via allopregnanolone-dependent mechanisms mediated by the pineal gland.

I think that the revised manuscript has been improved by the revisions, and am satisfied with it. I recommend publication.

In particular, I welcome the in-depth lobule-specific analyses that the authors have added, which shows the specificity of the effect to anterior lobules (which are closest to the pineal gland). Given recent interest in lobule-specific changes in the cerebellum, this adds to the richness of our understanding of region-specific Purkinje cell regulation.

I am also extremely pleased and impressed that the authors repeated their work in female chicks, and are thus able to report that the effects they report are observed in both sexes. I thank the authors for taking this concern seriously and for putting in the work to address it.

I think several of the changes made, including to analyses, representation of data, and to writing (e.g. removing difficult-to-understand abbreviations) have greatly strengthened the paper as well.

Reviewer #2:

The authors have addressed all comments regarding data presentation and analysis.

However, the Introduction of the revised manuscript is still illogical, with several statements that do not fit well. The meaning behind the third sentence ("It is not surprising that.…") is unclear. This sentence should be deleted. The reference to weight gain does not fit with the theme of development/maturation of the brain. Fagiolini et al., 2009, refer to the influence of stress, acting via the placenta, on the epigenome of the embryo. The notion of "environment" in this case is different from a stimulus that acts directly on a sensory system of the post-hatch/post-natal animal.

The Introduction would be clearer if the authors define what is meant by "development" in this manuscript. The experiments focus on post-hatch chicks, hence this should be emphasized. The statements in the Discussion on the effects of inappropriate lighting on post-natal brain (–Discussion, third paragraph) could be placed in the Introduction, to clarify the focus of the manuscript.

In several instances, the authors use the phrase "significantly increased" or "significantly decreased". It would be clearer to state "increased (p = x, or effect size = x)" or "decreased (effect size =).

Reviewer #3:

The authors have addressed the reviewers' suggestions satisfactorily. The data and statistics are now presented in a rigorous manner and the results of the study are compelling. That the decrease in Purkinje cell number persists in adulthood is significant. There are a few points which the authors must rectify before the manuscript can be accepted:

1) Figure 5C and D: Is Control versus Px+ALLO or Px+PACAP statistically significant? Do ALLO or PACAP treatments completely or partially rescue Caspase expression and Purkinje neuron survival?

2) The authors propose a model where PACAP expressed by Purkinje neurons binds to PAC1 receptors either in an autocrine or paracrine fashion. However, PACAP immunoreactivity is seen outside of Purkinje neurons too in the molecular layer (Figure 5E). Further PAC1 immunoreactivity is seen in the granule cell layer also (Figure 5F). Given these, one cannot be sure of either the origin of PACAP or its targets.

3) Subsection “Quantification of the number and dendritic length of Purkinje cells” and Figure 2—figure supplement 2: Molecular layer thickness cannot be equated to Purkinje neuron dendritic length.

---

## [Author Response]

Essential revisions:1) The extent of Purkinje neuron loss due to light-at-night in the current study seems to be small (Figure 1H). As it stands, it looks like there is a clear effect on pineal ALLO synthesis, but only a marginal effect on Purkinje cell survival. Small changes are not necessarily unimportant as even a decrease of 10% of cerebellar Purkinje cells could still have a big impact on cerebellar function. This raises two questions: one, if there are lobule-specific differences in Purkinje cell loss as seen in Haraguchi et al., 2012 with pinealectomy, and two if these small changes persist into adulthood and lead to cerebellar deficits. To delineate if there are lobule-specific differences, authors should provide lobule-specific counts of PNs after light-at-night. Such lobule-specific differences could potentially be quite interesting in terms of light regulation of Purkinje neuron survival. To see if these changes persist into adulthood, authors should look at counts at fully adult stages as well.

Thank you for this insightful comment. We agree with the reviewer that the decrease in Purkinje cell numbers in our study is small to lead to a change in cerebellar functions related to movement coordination, motor learning, or timing, etc. We think that the decrease of Purkinje cell number by light-at-night affects cerebellar non-motor functions but not motor functions. The cerebellum is also involved in autism spectrum disorder. Thus, we added several sentences in the Discussion section as follows:

“Circadian rhythms are present in almost all plants and animals. […] Further studies are needed to investigate the connection between a decrease in Purkinje cells caused by light-at-night and autism spectrum disorder.”

In addition, according to the reviewers' comments, we have provided lobule-specific counts of Purkinje cell numbers after light-at-night (please see the new Figures 2, 3, 5-7). Furthermore, the following information is added in the Materials and methods section as follows: “The number of calbindin-immunoreactive cell bodies of Purkinje cells was counted in each lobule at P21 or P70. […] The number of Purkinje cells in the vermis area was calculated from 10 sections (20-μm thickness, every three sections in the vermis area) per animal.” and “Parasagittal cerebellar sections of chicks at P7 were examined by immunostaining with an antibody against cleaved caspase-3 (Asp175; #9661; Cell Signaling Technology, Danvers, MA) to detect apoptotic cells immunoreactive to the active form of caspase-3 as described previously (Haraguchi et al., 2012a). The number of cleaved caspase-3-positive Purkinje cells in the vermis area was calculated from 10 sections (20-μm thickness, every three sections in the vermis area) per animal.”

Furthermore, we have performed an additional experiment to investigate whether the decrease in Purkinje cell numbers by light-at-night during the posthatch period persists into adulthood. In adulthood, Purkinje cell numbers were also decreased in lobules III, IV, and V by light-at-night. (Please see the new Figure 2G).

2) Data presentation: Throughout the manuscript data are presented as bar graphs showing mean+/- sem. It is now standard practice to show the scatter of the data with a suitable measure of central tendency depending on the shape of the distribution. Further, data in Figure 1G, H (and elsewhere in the paper, e.g. Figure 3B-E) are represented as relative to controls. It would be informative to have an indication of the actual number of cells. If necessary, this can be normalized to the total number of cells (or total fluorescence of a nuclear indicator) in each section.

Based on this insightful comment, we have revised our graphs. In this revised version we show the scatter of the data with a suitable measure of central tendency, depending on the shape of the distribution throughout the manuscript (please see the new figures). In addition, we have provided lobule-specific counts of Purkinje cell numbers (please see the new Figures 2, 3, 5-7).

3) The statistical analysis of the data is insufficient. It is not clear if the data are normal and normality tests were done, but a one-way ANOVA has been used. The authors should not take p<0.05 to indicate that their hypothesis is true. The statement by APA (Wasserstein, R. L. & Lazar, N. A. The ASA's Statement on p-Values: Context, Process, and Purpose. The American Statistician 70, 129-133 (2016)) should be consulted as to what the p-value can and cannot be used for. It is suggested that the authors use Gardner-Altman estimation plots (eg see estimationstats.com), and provide an indication of the size of the effect of the different treatments.

Based on this insightful comment, we have conducted several experiments to determine the number of *n*. Subsequently, we have showed the scatter of the data with a suitable measure of central tendency depending on the shape of the distribution throughout the manuscript (please see the new Figures). In addition, we have analyzed the effect size of data by estimation statistics. To analyze two groups, we used a Gardner-Altman estimation plots. To analyze a single control and each of the intervention groups, we used a shared-control estimation plot, instead of Gardner-Altman estimation plots (please see the new Figure 2—figure supplement 3).

We have added the following sentences regarding statistics in the Materials and methods section as follows: “Statistical analysis. GraphPad Prism (GraphPad) and Estimation statistics (http://www.estimationstats.com) were used for statistical analysis. […] P values of the relevant post hoc multiple comparisons are shown in the figures. The statistical significance cutoff was set at P < 0.05.”

4) The authors should include example representative images for caspase-3 positive cells and Purkinje cells in Figure 2, so that the reader can evaluate these findings themselves.

Based on this insightful comment, we have added example representative images for caspase-3 positive cells and Purkinje cells (please see the new Figure 2D and F).

5) Why were male chicks used exclusively? In previous studies by these authors, results were shown to be similar in both male and female chicks. These results will be stronger if they are confirmed in females. At minimum, a justification for the exclusive use of male animals should be provided.

Thank you for this insightful comment. We agree with the importance of females, and have performed an additional experiment to investigate the effects of light-at-night on female chicks. Purkinje cell numbers were also decreased in female chicks by light-at-night (please see the new Figure 2—figure supplement 1).

In addition, we have added the following information in the Materials and methods section as follows: “Animals. Domestic chickens (*Gallus gallus*) of both sexes at various ages were used in this study. Chicks were incubated under either a 12 h light–12 h dark (LD) cycle, constant light (LL), or a 12 h light–12 h dark cycle followed by exposure to light for 1 h from ZT14 to ZT15 (light-at-night) with the light provided by white fluorescent lamps.”

6) Why has this particular light-at-night paradigm been used? It would be useful if the paradigm were described in the Materials and methods section (it is currently not mentioned there at all) and a brief discussion of the generalizability of this light-at-night paradigm could be included in the Discussion.

Thank you very much for this insightful comment. Based on this comment, we have revised and added several sentences in the Discussion section as follows:

“Circadian rhythms are present in almost all plants and animals. […] Further studies are needed to investigate the connection between a decrease in Purkinje cells caused by light-at-night and autism spectrum disorder.”

7) At what time was the ALLO injection given? The methods section mentions that vehicle only injections were done but it is not clear if the 'control' is vehicle-only or uninjected animals. Both conditions must be shown, with raw cell counts.

Thank you very much for this insightful comment. Based on this comment, we have added the following sentences in the Materials and methods section as follows:

“Quantification of the number and dendritic length of Purkinje cells. […] Control animals were treated with an equal volume of vehicle-only.”

In addition, according to important comments, we have provided lobule-specific counts of Purkinje cell numbers after light-at-night (please see the new Figures 2, 3, 5-7).

8) The authors go into some detail delineating the timeline of the action of light at night during development. However, it is hard to understand how this is believed to correspond to human (or mammalian) development, so it would be useful to discuss this more directly in the Discussion.

Thank you very much for this important comment to improve the manuscript. According to this comment, we have revised and added several sentences in the Discussion section as follows:

“Circadian rhythms are present in almost all plants and animals. […] Further studies are needed to investigate the connection between a decrease in Purkinje cells caused by light-at-night and autism spectrum disorder.”

9) The second and third sentences of the Introductory paragraph could use some rewriting as they are only loosely connected to the rest of the text. "In particular, light conditions affect development during early life; therefore, it is not surprising that life evolved on this planet") is not logically sound.

Thank you very much for this insightful comment. Based on your comment, we have revised several sentences in the Introduction section as follows: “Environmental stimuli (e.g., light–dark cycle, temperature, or nutrition) influence the development of plants, animals, and humans. […] Several studies have reported that circadian disruption by light-at-night affects weight gain in vertebrates during early neonatal or posthatch life”

[Editors' note: further revisions were requested prior to acceptance, as described below.]

The reviewers were in general happy with the revised manuscript and felt that the revisions made it much stronger. However, there are a few points which need to be rectified before the manuscript can be accepted. Please see these below in the comments section and address them at the earliest.Reviewer #2:The authors have addressed all comments regarding data presentation and analysis.However, the Introduction of the revised manuscript is still illogical, with several statements that do not fit well. The meaning behind the third sentence ("It is not surprising that.…..") is unclear. This sentence should be deleted.

Thank you very much for your comment. Based on this comment, we have deleted the aforementioned sentence from the Introduction.

The reference to weight gain does not fit with the theme of development/maturation of the brain. Fagiolini et al., 2009, refer to the influence of stress, acting via the placenta, on the epigenome of the embryo. The notion of "environment" in this case is different from a stimulus that acts directly on a sensory system of the post-hatch/post-natal animal.

Thank you very much for this insightful comment. Based on this comment, we have deleted the relevant sentence from the Introduction.

According to this comment, we have revised the Introduction as follows:

“Environmental stimuli (e.g., light–dark cycle, temperature, or nutrition) influence the development of plants, animals, and humans. […] However, little is known about the molecular mechanisms that control how environmental light conditions affect brain development.”

The Introduction would be clearer if the authors define what is meant by "development" in this manuscript. The experiments focus on post-hatch chicks, hence this should be emphasized. The statements in the discussion on the effects of inappropriate lighting on post-natal brain (Discussion, third paragraph) could be placed in the Introduction, to clarify the focus of the manuscript.

Thank you very much for this insightful comment and kind suggestion. Based on this comment, we have changed several sentences regarding the effects of inappropriate lighting on the post-natal brain in the Introduction as follows:

“Multiple studies have suggested a link between light-at-night-induced circadian disruption and disruption of the neuroendocrine system (Fonken and Nelson, 2014; Fonken et al., 2010). […] However, the molecular mechanisms that modulate the expression of hormones depending on light conditions during early life are still incompletely understood.”

In several instances, the authors use the phrase "significantly increased" or "significantly decreased". It would be clearer to state "increased (p = x, or effect size = x)" or "decreased (effect size =).

Based on this comment, we have changed “significantly increased” to “increased (P=X)” or “significantly decreased” to “decreased (P=X)” in the re-revised manuscript.

Reviewer #3:The authors have addressed the reviewers' suggestions satisfactorily. The data and statistics are now presented in a rigorous manner and the results of the study are compelling. That the decrease in Purkinje cell number persists in adulthood is significant. There are a few points which the authors must rectify before the manuscript can be accepted:1) Figure 5C and D: Is Control versus Px+ALLO or Px+PACAP statistically significant? Do ALLO or PACAP treatments completely or partially rescue Caspase expression and Purkinje neuron survival?

In Figure 5C, we did not detect any statistically significant differences between Control and Px+ALLO (P>0.9999) or Px+PACAP (P=0.8672).

In Figure 5D, we detected a statistically significant difference between Control and Px+ALLO (P=0.0002), but not between Control and Px+PACAP (P=0.0533).

Thus, our results indicated that ALLO treatment partially rescued caspase-3 expression and Purkinje neuron survival. In addition, PACAP treatment completely rescued both caspase-3 expression and Purkinje neuron survival. To reflect these results in the manuscript, we have revised Figure 5D.

2) The authors propose a model where PACAP expressed by Purkinje neurons binds to PAC1 receptors either in an autocrine or paracrine fashion. However, PACAP immunoreactivity is seen outside of Purkinje neurons too in the molecular layer (Figure 5E). Further PAC1 immunoreactivity is seen in the granule cell layer also (Figure 5F). Given these, one cannot be sure of either the origin of PACAP or its targets.

Thank you very much for this insightful comment. Based on this comment, we have added several sentences in the Discussion as follows:

“In the developing cerebellum, PACAP was expressed not only by Purkinje cells but also by the cells of the molecular layer. […] However, the effect of PACAP on granule cells is unclear in the developing cerebellum of chicks, although the neuroprotective effects of PACAP on cerebellar granule cells have been reported in mammals (Vaudry et al., 2000; Vaudry et al., 2003)”.

3) Subsection “Quantification of the number and dendritic length of Purkinje cells” and Figure 2—figure supplement 2: Molecular layer thickness cannot be equated to Purkinje neuron dendritic length.

Thank you very much for this insightful comment. Based on this comment, we have changed “Quantification of the number and dendritic length of Purkinje cells.” to “Quantification of the number of Purkinje cells and the thickness of the molecular layer.”

We have also changed “Effect of pineal ALLO on the thickness of the molecular layer of Purkinje cells.” to “Effect of pineal ALLO on the thickness of the molecular layer.”

Moreover, we have changed “Dendritic length of Purkinje cells in lobule IV at P21” to “The thickness of the molecular layer in lobule IV at P21”